# Distributable, metabolic PET reporting of tuberculosis

Tuberculosis remains a large global disease burden for which treatment regimens are protracted and monitoring of disease activity difficult. Existing detection methods rely almost exclusively on bacterial culture from sputum which limits sampling to organisms on the pulmonary surface. Advances in monitoring tuberculous lesions have utilized the common glucoside [18F]FDG, yet lack specificity to the causative pathogen *Mycobacterium tuberculosis* (*Mtb*) and so do not directly correlate with pathogen viability. Here we show that a close mimic that is also positron-emitting of the non-mammalian *Mtb* disaccharide trehalose – 2-[18F]fluoro-2-deoxytrehalose ([18F]FDT) – is a mechanism-based reporter of Mycobacteria-selective enzyme activity in vivo. Use of [18F]FDT in the imaging of *Mtb* in diverse models of disease, including non-human primates, successfully co-opts *Mtb*-mediated processing of trehalose to allow the specific imaging of TB-associated lesions and to monitor the effects of treatment. A pyrogen-free, direct enzyme-catalyzed process for its radiochemical synthesis allows the ready production of [18F]FDT from the most globally-abundant organic 18F-containing molecule, [18F]FDG. The full, pre-clinical validation of both production method and [18F]FDT now creates a new, bacterium-selective candidate for clinical evaluation. We anticipate that this distributable technology to generate clinical-grade [18F]FDT directly from the widely-available clinical reagent [18F]FDG, without need for either custom-made radioisotope generation or specialist chemical methods and/or facilities, could now usher in global, democratized access to a TB-specific PET tracer.

Tuberculosis (TB) caused by *Mycobacterium tuberculosis* (*Mtb*) still remains a serious global health challenge causing an estimated 1.3 million deaths worldwide in 2022[1]. Prompt, short-term diagnoses of TB are crucial for public health infection control measures, as well as for ensuring appropriate treatment for infected patients and controls[2]. The global number newly diagnosed of 7.5 million is the largest since the WHO began monitoring in 1995[1]. Additionally, long-term accurate monitoring of chronic disease burden and the effectiveness of treatment is critically important in trials of new antitubercular agents and regimens. Sensitive and TB-specific reporters with the potential for ready democratization are therefore urgently required to address the development of new

antituberculosis agents and regimens with the potential to shorten the duration of therapy.

Positron emission tomography (PET) integrated with computed tomography (CT) now routinely provides a prominent method in some (and increasing[3]) national healthcare systems for noninvasively imaging the whole body whilst diagnosing, staging and assessing response to therapy in diseases such as cancer and inflammation. The analysis[4] and internationally-agreed monitoring of access to PET-CT (e.g. by the IAEA Medical imAGIng and Nuclear mEdicine global resources database, IMAGINE[5]) helps, in part, to drive global equity of access to such diagnostic imaging. However, this would be importantly aided by development of (i) novel reporters specific to other (e.g.

✉ e-mail: CBARRY@niaid.nih.gov; Ben.Davis@rfi.ac.uk

communicable) diseases of similar, or even greater, relevance to the developing world, such as TB, and (ii) strategies and methods for their ready, distributed implementation.

Existing metabolic PET probes allow pharmacological, immunologic and microbiological aspects of TB lesions to be correlated with anatomic information derived from PET/CT[6,7]. In particular, 2-[18F] fluoro-2-deoxy-D-glucose ([18F]FDG, often shortened to FDG, Fig. 1), as the most widespread, organic [18F]-containing probe, has allowed useful diagnosis and monitoring of the response to treatment of TB[8,9]. Specifically, the high metabolic uptake of [18F]FDG into the host cells around pulmonary and extra-pulmonary lesions of active TB can allow PET/CT imaging of TB granulomas by inference and so attempts to indirectly assess disease extent and progression or resolution[10]. [18F] FDG PET exploits enhanced glucose uptake into the anaerobic glycolytic pathway both in tumor cells and in immune cells – [18F]FDG PET in imaging of infections relies therefore typically on enhanced glucose uptake of inflamed cells as a result of associated respiratory burst. However, by virtue of this physiological mechanism, [18F]FDG is also taken up and retained by any metabolically-active tissue and so, as a generic marker of more active metabolism, has a limitation in its lack of specificity and inability to clearly distinguish granulomatous TB disease from other inflammatory conditions, including cancer. This 'inferred imaging' mode therefore frequently gives rise to false-positive diagnosis in patients evaluated for active TB[11–13]. This

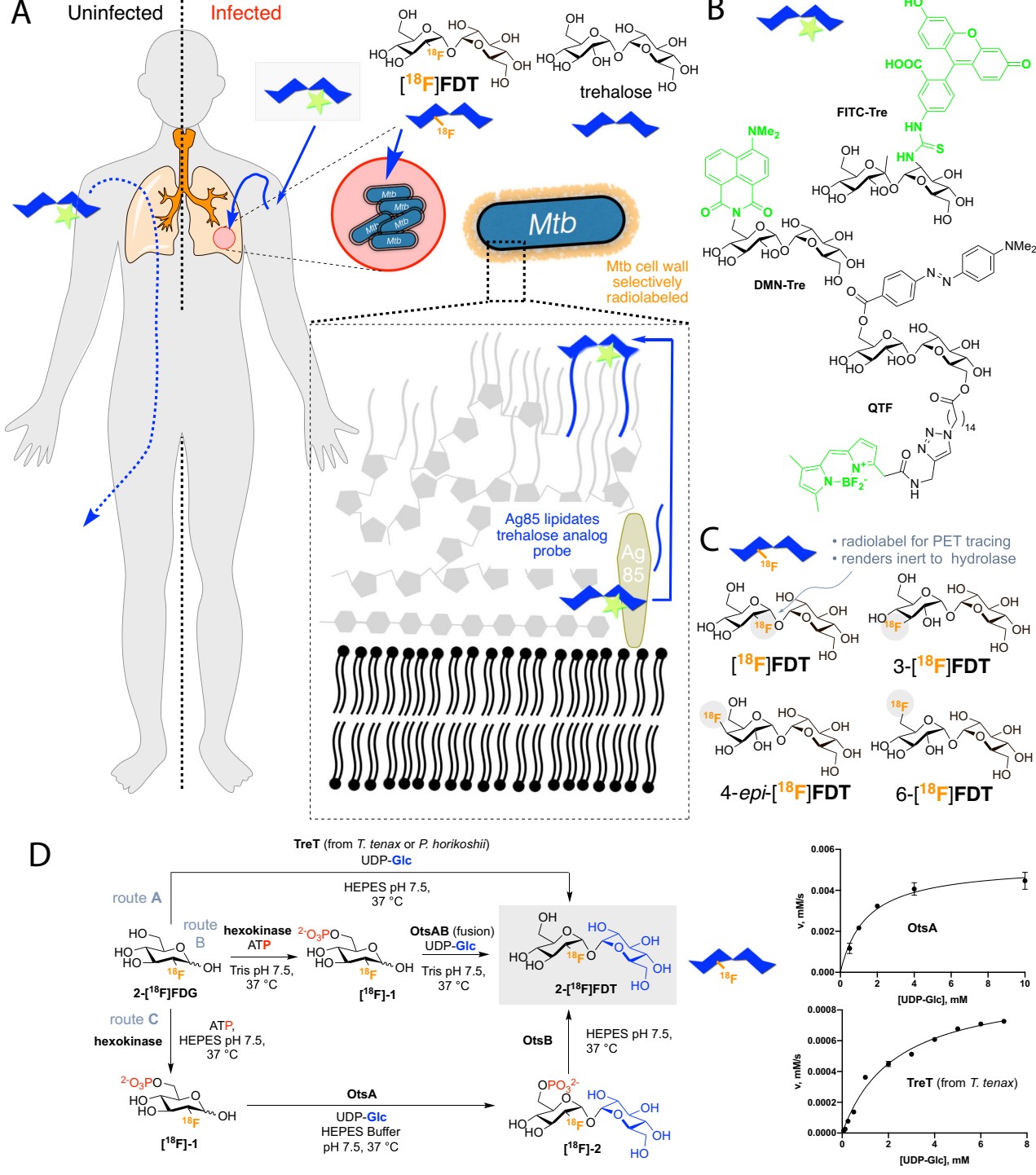

**Fig. 1 | A Strategy for Non-Invasive Imaging Reporters using Trehalose-based, TB-specific Probes Derived Directly from [18F]FDG. A** Trehalose (blue) in *Mtb* is found in the outer portion of the mycobacterial cell envelope as its corresponding mycolate glycolipids. The biosynthesis of trehalose mycolates (lipidation) is catalyzed specifically in *Mtb* by abundant membrane-associated Antigen 85 (Ag85a, Ag85b and Ag85c) enzymes. The lack of naturally occurring trehalose in mammalian hosts as well as the uptake of exogenous trehalose by *Mtb* suggests that it could function as both a highly specific and sensitive probe, allowing here the development of an in vivo TB-specific, PET-radiotracer analog [18F]FDT that selectively labels lesions (red) in infected organisms (right). Uninfected organisms (left) do not process trehalose and so probe is not retained. **B** Prior work has established that Ag85s are sufficiently plastic in their substrate scope that they can process, for example, fluorescent analogs of trehalose, allowing them to be metabolically incorporated into the mycobacterial outer membrane in vitro for labeling. Fluorescence-based methods are not yet amenable to effective, non-invasive imaging in vivo. **C** Four 18F-labeled variants of trehalose were tested in which each of the available hydroxyls (OH−2, 3, 4, 6) were converted in turn to 18F. **D** Enzymatic synthesis of [18F]FDT. Three parallel routes (A, B and C) were evaluated for

efficiency, rate and yield. Route A: TreT-mediated synthesis was evaluated using enzymes from two different sources (from *T. tenax* or *P. horikoshii*). Both proved functional but gave lower turnover under a range of conditions (see pseudo-single substrate plot for TreT (from *T. tenax*), lower right – See Fig. S1 and Source Data File for further details of kinetics). [Reagents and Conditions: 50 mM HEPES, 100 mM NaCl and 10 mM MgCl$_2$, pH 7.5]. Route B: OtsAB-fusion enzyme-mediated synthesis was explored. An OtsAB fusion protein was constructed but proved to be more difficult to express and less stable under typical reaction conditions [Reagents and Conditions: 50 mM HEPES, 100 mM NaCl and 10 mM MgCl$_2$, pH 7.5 C] Route C: Although a three-step, three-enzyme route, Route C proved to be more flexible and reliable. The selectivity of the biocatalysts allowed this to be performed in a convenient one-pot manner. The greater stability of enzyme components and the ability to vary catalyst amounts to control flux led to its choice over Route B. The higher turnovers and efficiencies led to its choice over Route A (see pseudo-single substrate plot for OtsA, upper right – See Fig. S1 and Source Data File for further details of kinetics). [Reagents and Conditions: 50 mM HEPES, 100 mM NaCl and 10 mM MgCl$_2$, pH 7.5.]. The data points are average values from three replicates with error bars ± SD(n = 3).

diagnostic confusion when using [18F]FDG has been further exacerbated by the wide-spread emergence of lung pathology in COVID-19 patients[14].

With the aim of improving detection specificity, other [18F]-labeled PET tracers have been investigated for imaging of TB such as 2-[18F] fluoroisonicotinic acid hydrazide[15] and [18F]deoxy-fluoro-L-thymidine (FLT)[16]. However, the sensitivity and specificity of these radiotracers is either only similar to (or worse than) [18F]FDG and so can only help to distinguish TB from malignancy when also combined with [18F]FDG. Moreover, neither probe are yet readily accessible from common precursors; both require specialist generation of radioisotopes (e.g. [18F]fluoride via cyclotron-mediated proton (1H) irradiation of H$_2$18O) combined with specific chemical technologies (e.g. appropriate synthetic laboratories and methods).

Trehalose, a nonreducing disaccharide in which the two glucose units are linked in an α,α−1,1-glycosidic linkage (Fig. 1A) has, in the last decade, gathered interest as an agent for selectively imaging TB. Trehalose is found in *Mtb*, especially in the outer portion of the mycobacterial cell envelope – primarily as its corresponding glycolipids trehalose monomycolate (TMM) and trehalose dimycolate (TDM). These glycolipids appear to play a critical structural, and perhaps pathogenic role, as an essential cell wall component[17,18], as has long been noted (TDM was first referred to as 'cord-factor' in the 1950s[19]). The biosynthesis of trehalose mycolates is catalyzed in *Mtb* by a family of abundant TB-specific, membrane-associated enzymes – the Antigen 85 proteins (Ag85a, Ag85b and Ag85c)[20].

The lack of naturally-occurring trehalose in mammalian hosts as well as the uptake of exogenous trehalose by *Mtb*[21] has suggested that trehalose-based probes could function as both highly specific and sensitive reporters (Fig. 1A). Prior work has established that the Ag85 family enzymes (Ag85s) are somewhat plastic in the substrates that they can process – not only can exogenous trehalose be processed by Ag85s but also analogs of trehalose. In this way, they can be metabolically incorporated into the mycobacterial outer membrane[21]. This allowed fluorescent labeling and monitoring of *Mtb* in vitro, not only in bacilli but also within infected mammalian macrophages using reagents such as FITC-Tre (Fig. 1B)[21]. More recently, elegant design has created yet more powerful fluorescent variants, such as the fluorogenic QTF[22], and FRET-TDM[23] and solvatochromic DMN-Tre[24] (Fig. 1B), some of which enable detection of *Mtb* detection in sputum[24], as well as photosensitized variants for possible photodynamic therapy[25]. However, fluorescence-based methods are not yet amenable to effective, non-invasive clinical in vivo imaging and, to our knowledge, no trehalose-based probe or method has yet demonstrated efficacy in vivo. Here we show that one of the simplest 18F-analogs of trehalose, 2-[18F]fluoro-2-deoxy-

trehalose [18F]FDT can be generated as an in vivo TB-reporter using one-pot, automatable, pyrogen-free, chemoenzymatic synthesis from readily-available [18F]FDG in a validated manner that does not require specialist expertize (Fig. 1C); initial aspects of these methods were disclosed previously[26,27]. Toxicological and preclinical testing in diverse species shows that this now creates access to a safe probe that is effective and selective in multiple preclinical models both to visualize TB lesions and to monitor their treatment non-invasively. [18F]FDT is therefore now suggested as a suitable candidate for clinical use, enabled by a distributable synthetic strategy that could help lower global health technology imbalances.

## Results

### Comparison of four candidate 18F-fluoro-deoxy analogs of trehalose as probes identifies [18F]FDT as a likely radiotracer reporter of TB

To test the effectiveness of different minimal alterations at different sites in the trehalose scaffold (Fig. 1C), we synthesized four radiolabeled 18F-fluoro-analogs of trehalose. We aimed to use complementary, straightforward chemical and enzymatic approaches that were initially developed via semi-analogous 'cold' routes (see Figs. S1–S4). In each fluoro-trehalose analog a single hydroxyl at each of the four different sites in the trehalose scaffold was replaced by a fluoro group – specifically at positions 2-, 3-, 4 and 6- giving the corresponding deoxy-[18F]fluoro analogs: [18F]FDT, 3-[18F]FDT, 4-*epi*-[18F]FDT (chosen as an epimeric target due to synthetic expediency) and 6-[18F]FDT (Fig. 1C).

In our design (Fig. 1C) of these minimally-altered trehalose analogs, we considered not only tolerance for the PET-tracing label (due to minimal structural changes) for co-opted processing by *Mtb* but also we were mindful of the potential inhibitory properties that such compounds might have upon putative host degrading enzymes[28] – 2-fluoro-2-deoxy-sugar substrates, for example, have been elegantly shown to act as irreversible, mechanism-based inhibitors of corresponding sugar hydrolases[29].

Specifically, in brief, 4-*epi*-[18F]FDT and 6-[18F]FDT novel radiochemical syntheses used nucleophilic fluorination protocols that generated product from corresponding triflate precursors in good yields and with high purity (Figs. S3A, B, S4). As such, these synthetic approaches were strategically similar to typical syntheses of, for example, commercially-available [18F]FDG, where aqueous [18F]-fluoride is used as the source of 18F[30]. Protocols for [18F]FDT, 3-[18F]FDT utilized enzymatic methods (see the Supplementary Methods and following sections for details).

Whilst the syntheses of [18F]FDT, 4-*epi*-[18F]FDT and 6-[18F]FDT proved sufficiently rapid (complete within one half-life of 18F, t$_{1/}$

2 ~ 110 min), that of 3-[18F]FDT was slower (consistent with results for 'cold' analogs) and less efficient (3-[18F]FDT, 10% conversion (n = 2), 60 min to 4 h). Moreover, whilst 4-*epi*-[18F]FDT and 6-[18F]FDT could be prepared from [18F]-fluoride used at a later stage (substitution then deprotection, see Figs. S2–S4), 3-[18F]FDT required the prior preparation of deprotected 3-[18F]fluoro-3-deoxy-glucose before enzymatic processing, which was itself relatively inefficient[31]. The particular difficulties in the synthesis of 3-[18F]FDT (length of reaction times, poor conversion and need for early use of radionuclide) led to low specific activities and so this candidate was dismissed at an early stage.

The metabolic stabilities of the remaining three analogs, [18F]FDT, 4-*epi*-[18F]FDT and 6-[18F]FDT were tested. Although, human trehalose-degrading trehalase activity is low and restricted primarily to the brush-border of the kidney where it is thought to be GPI-anchored[28,32], it can be elevated in some diseases[28], therefore we also tested degradation of the analogs in vitro using high concentrations of mammalian (porcine) trehalase (0.3 units/mL, 10 mM substrate, Fig. S5). Under conditions that led to complete degradation of trehalose, 'cold' analogs [19F]FDT, 4-*epi*-[19F]FDT showed little or no degradation, whereas 6-[19F]FDT showed low levels (14%).

Finally, putative degradation was also probed directly in vivo using both *Mtb*-infected and naïve rabbits. Consistent with the in vitro observations, radiotracer 6-[18F]FDT was less stable than [18F]FDT and 4-*epi*-[18F]FDT; 6-[18F]FDT was entirely metabolized to 6-[18F]FDG within 90 min post-injection, (see Fig. S3D). Whilst 4-*epi*-[18F]FDT proved more stable, some metabolism (albeit slower) was observed (30% after 3 h (Fig. S3C). However, consistent with design (Fig. 1c), [18F]FDT showed little apparent degradation (see below).

Taken together, the rapid degradation of 6-[18F]FDT and both the partial degradation and less efficient (lower radiochemical yield (RCY) and a need for more stringent purification) synthesis of 4-*epi*-[18F]FDT led to their dismissal as candidate tracers; [18F]FDT was evaluated further.

## Comparison of synthetic routes suggests a flexible one-pot, biocatalytic route to [18F]FDT

Three parallel strategies were envisaged for the synthesis of [18F]FDT (Fig. 1D) and these were tested in the synthesis of [19F]FDT. We aimed from the outset to devise routes that would ultimately utilize [18F]FDG, as this is the most readily-available organic source of 18F, even to those without specialist sources of 18F (e.g. cyclotron facilities). Global access via commercial sources would therefore, in principle, allow the development of synthesis that could be implemented in many locations.

[18F]FDG is typically supplied in aqueous solution and so all routes adopted a protecting-group-free approach that would not only obviate the need for additional (e.g. protection-deprotection) steps, but, through the use of biocatalysts, allow the use of such solutions directly. Given the often highly (chemo-, regio- and stereo-) selective nature of biocatalysts we envisaged that all three routes would have potential for application in a one-pot operation for ease-of-use. Four biocatalyst systems were evaluated for mediating the critical formation of the α,α−1,1-*bis*-glycosidic linkage that is found in [18F]FDT, due to the nature of its dual *cis*−1,2-linkage[33]. This is a particularly difficult linkage to form and control selectively using chemical methods, further highlighting another potential advantage of the use of biocatalysis. Two homologs of the, so-called, trehalose glycosyltransferring synthase enzyme, TreT[34], (from *T. tenax*[35–38] and *P. horikoshii*[39]) were tested via Route A; and two variant constructs containing the trehalose 6-phosphate synthase domain OtsA[40], one in Route B as a fusion to the dephosphorylating enzyme OtsB and one as a single enzyme construct via Route C.

First, for initial syntheses, all enzymes were expressed in and purified from *E. coli*. The yield for OtsA enzyme as an individual construct was superior to the OtsAB fusion construct and/or TreT homologs. OtsB from *E. coli* did not express in the soluble fraction but could either be purified from inclusion bodies or its solubility improved through the addition of a fused-*N*-terminal-MBP-His-tag (see Supplementary Methods and Figs. S6–S8).

Next, we determined kinetic parameters of the various enzyme systems (Fig. S1 and Table 1) using a variety of complementary (continuous or endpoint; coupled-enzyme[41], spectrophotometric, HPLC or 1H NMR) methods to monitor substrate consumptions and/or product formation, according to conditions. Again, individual OtsA and OtsB constructs proved superior catalysts (Table 1) to the corresponding OtsAB fusion and, noting the limits on comparisons, to the two TreT enzymes, with OtsA catalytic efficiencies some >2-fold to 4-fold higher, as judged by $k_{cat}/K_M$. Here, it should be noted, we used endogenous Glc or Glc6P as acceptor substrate rather than the deoxyfluoro sugars and so these parameters should be interpreted as a proxy – direct comparisons with the kinetics used for eventual FDT synthesis therefore cannot be made (see also below). The relative efficiency of OtsB was higher ( ~ 4-fold to 10-fold) than OtsA allowing its use in lower concentration, further highlighting the advantage of this separate construct (see below).

These catalytic efficiencies had a direct bearing on synthetic utility. In comparative 10 mM reactions: via Route A (Fig. 1D) TreT from *P. horikoshii* performed better (40% conversion over 2 h) than the TreT from *T. tenax* (10% conversion over 2 h); *P. horikoshii* TreT also displayed reversibility under certain conditions that would contaminate product FDT by converting it back to starting FDG. Via Route B using OtsAB, synthesis failed (even over prolonged periods). By contrast, via Route C using OtsA and OtsB full conversion was observed within 45 min (Fig. S9). Following these evaluations Route C (Fig. 1) was selected over both Routes A and B. It is important to note that these comparisons were conducted under specific conditions that may not have been optimized for the given systems in our hands[37,38].

## Creation of a pyrogen-free synthetic biocatalyst system enables a safe route to [18F]FDT

Pyrogen or endotoxin presence due to lipopolysaccharide contamination caused, for example, by carryover from *E. coli* production is

**Table 1 | Kinetic parameters of enzymes used in [19F]FDT synthesis**

| Enzyme | Substrate | $K_M$ (mM) | $k_{cat}$ (s⁻¹) | $k_{cat}/K_M$ (mM⁻¹s⁻¹) |
|---|---|---|---|---|
| OtsA [a] | UDP-Glc | 1.7 ± 0.3 | 34 ± 1 | 20 |
| OtsA [a] | G6P | 7.3 ± 0.6 | 34 ± 1 | 4.7 |
| TreT *P. horikoshii* [b] | UDP-Glc | 4.6 ± 0.8 | 25 ± 2 | 5.5 |
| TreT *P. horikoshii* [b] | Glc | 8.1 ± 0.6 | 13.6 ± 0.4 | 1.7 |
| TreT *T. tenax* [c] | UDP-Glc | 2.3 ± 0.7 | 13.5 ± 1.4 | 6.2 |
| TreT *T. tenax* [b] | Glc | 5.0 ± 1.3 | 10.3 ± 1.3 | 2.1 |
| OtsAB-fusion [d] | UDP-Glc | 7.2 ± 5.7 | - | - |
| OtsB [e] | T6P | 3.6 ± 0.8 | 286 ± 34 | 79 |
| OtsA[Sf] [b] | UDP-Glc | 1.4 ± 0.2 | 63.5 ± 2.5 | 45.3 |
| OtsA[Sf] [b] | G6P | 4.9 ± 0.5 | 57.5 ± 2.3 | 11.9 |
| OtsB[Sf] [e] | T6P | 2.8 ± 0.7 | 740 ± 96 | 264 |

[a] Taken from reference. [46]
[b] Determined via continuous, coupled-enzyme spectrophotometric assay[41] using decrease in NADH absorbance at 340 nm (ε340 = 6220 M⁻¹cm⁻¹) catalyzed by linked spectrophotometric assay at 25 °C.
[c] Determined via endpoint, HPLC method with fixed Glc concentration at 70 °C.
[d] Determined via endpoint, HPLC method with fixed Glc concentration at 37 °C.
[e] Determined using continuous 1H NMR assay at 37 °C. See Supplementary Methods for further details. OtsA[Sf] and OtsB[Sf] denote variants expressed from *Spodoptera frugiperda* rather than *E. coli*.

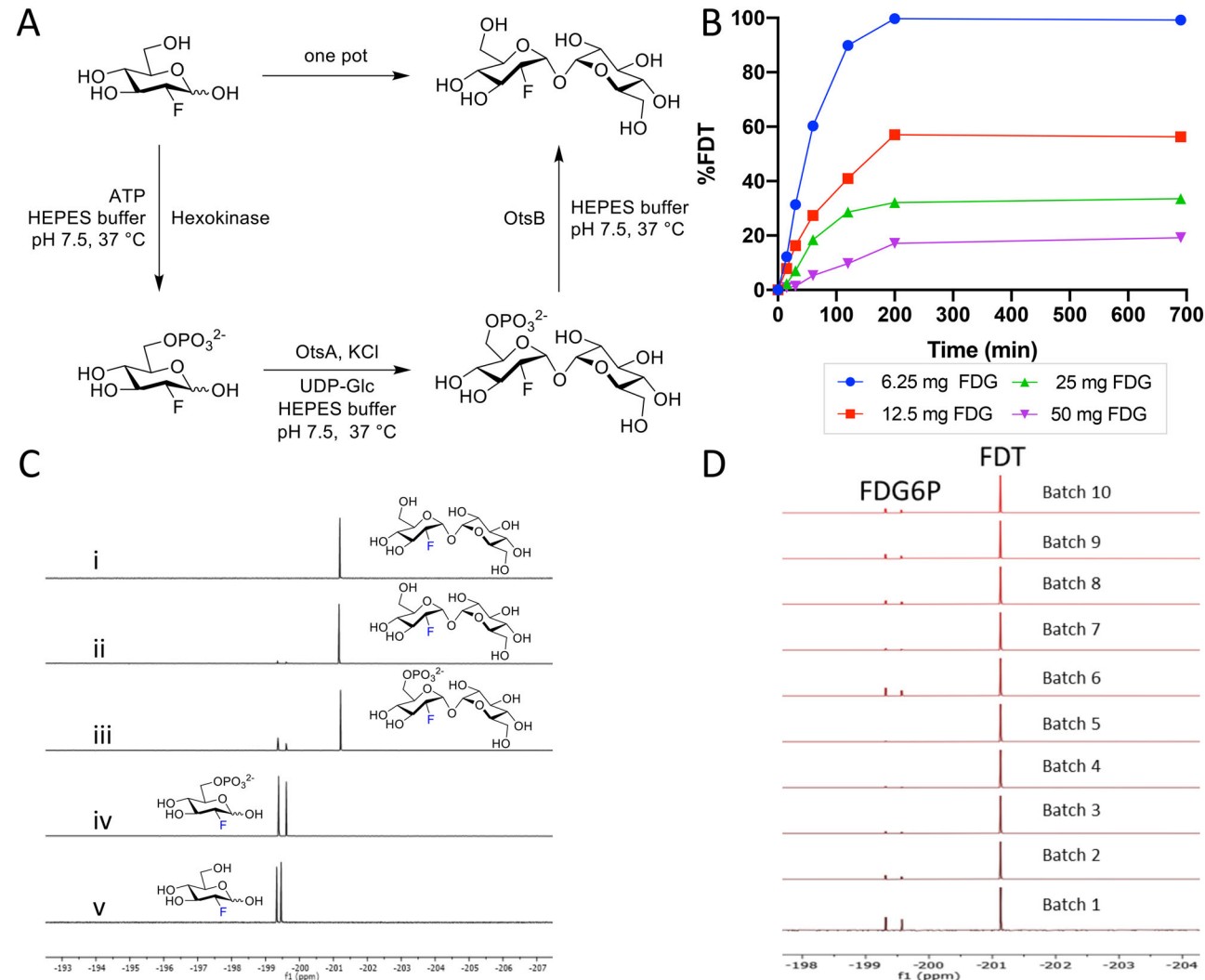

**Fig. 2 | Reaction Optimization, Scale-up and Batch Synthesis in the Development of an Efficient Scaleable One-Pot Synthesis. A** Chosen Route C (from Fig. 1) was tested in a one-pot format [¹⁹F]FDT using pyrogen-free enzymes OtsA$^{Sf}$ and OtsB$^{Sf}$. **B** Example reaction optimization with fixed donor sugar Glc-UDP [30 mM] at different acceptor substrate concentrations (9.08, 18.17, 36.34 and 72.68 mM) of [¹⁹F]FDG; reactions were monitored in real time by calibrated ¹⁹F NMR. Please also see **Source Data File**. **C** Direct reaction monitoring of the steps of one-pot, 3-step [¹⁹F]FDT synthesis from [¹⁹F]FDG by ¹⁹F-NMR. NMR spectra: **(i)** ¹⁹F NMR spectrum of

purified [¹⁹F]FDT; **(ii)** Crude ¹⁹F NMR spectrum of [¹⁹F]FDT with small amount of deoxy-fluoro-G6P (**[¹⁹F]-1**) still present; **(iii)** Crude ¹⁹F NMR spectrum of intermediates **[¹⁹F]-1** and **[¹⁹F]-2**; **(iv)** Crude ¹⁹F NMR spectrum of **[¹⁹F]-1** from [¹⁹F]FDG conversion; and **(v)** reference sample of starting material [¹⁹F]FDG.
**D** Representative, ¹⁹F-NMR spectra of crude reaction mixtures containing [¹⁹F]FDT from repeated batches using fresh enzyme (including from newly expressed preparations) and reactants synthesized by 3-enzymes-3-steps, one-pot syntheses, prior to purification.

prominent and of serious concern in any potential clinical method[42]. Whilst we were able to successfully utilize extensive, repeated cyclical purification coupled with pyrogen-monitoring using *Limulus* amoebocyte lysate activity measurements[43] to demonstrate low pyrogen production of [¹⁹F]FDT (Table S1), it has been argued that depyrogenation methods cannot be fully effective[42,44].

We therefore developed an expression system free of endotoxin based on the use of *Spodoptera frugiperda* (Sf9) cell culture. An appropriate sequence was designed based on the pFastbac vector system; the coding sequences of *E. coli* OtsA (CAA48913.1) and *E. coli* OtsB (CAA48912.1) were refactored according to *S. frugiperda* MNVP codon usage. Analysis of the resulting gene sequences for *E. coli* and *S. frugiperda* constructs confirmed essentially identical protein gene product primary sequence (Fig. S10). This allowed ready implementation of a protein production workflow Fig. S11) using infectious baculovirus generated through transposon-mediated insertion[45] (so-called Bac-to-Bac system). Sf9-culture resulted in good yields of proteins OtsA$^{Sf}$ and OtsB$^{Sf}$ (~ 20 mg/L of culture for OtsA$^{Sf}$; ~15 mg/L of

culture for OtsB$^{Sf}$, Fig. S12) that were essentially endotoxin-free (< 0.1 EU.mL⁻¹). This was comparable or superior to those found from *E. coli* expression of OtsA or OtsB and, indeed, of TreT and OtsAB fusion enzymes (both TreT enzymes ~2–3 mg/L, OtsAB fusion ~0.1 mg/L of culture). Both OtsA$^{Sf}$ and OtsB$^{Sf}$ proved stable up to 37 °C (Fig. S13). Moreover, the kinetic parameters of OtsA$^{Sf}$ and OtsB$^{Sf}$ (from Sf9 insect cells) also proved superior to those expressed from *E. coli*[46] (2-to-3-fold more efficient as judged by $k_{cat}/K_M$, Table 1). Finally and importantly, we were able to readily scale enzyme expression in Sf9 up to 10 L in one batch for both, thereby allowing corresponding production of >100 mg of either of the OtsA$^{Sf}$ and OtsB$^{Sf}$ enzymes.

### Development of an efficient, scaleable radiosynthesis provides [¹⁸F]FDT for pre-clinial testing

With a putative biocatalytic system in hand, we next sought to optimize its application in the synthesis of [¹⁹F]FDT as an isotopologue model for [¹⁸F]FDT (Fig. 2). Reaction parameters were varied (see Supplementary Methods), including substrate and enzyme

concentrations as well as duration of reactions. First, independently-variable, biocatalyst concentrations further verified the choice of flexible Route C (Fig. 2A) and allowed optimization of relative $[E]_0$ consistent with their kinetic parameters (Table 1) to allow maximal flux; this identified of 18.2 μM OtsA$^{Sf}$ and 6.1 μM OtsB$^{Sf}$ (Figs. S14, S15) as efficient. Second, variation of acceptor substrate [$^{19}$F]FDG concentration in the presence of fixed, maintained amounts of donor substrate UDP-Glc revealed more rapid reaction progress in the presence of equimolar or excess donor, consistent with both a highly efficient glycosylation reaction and also inhibition that could be minimized through control of conditions (Fig. 2B); variation of donor concentration under these constraints then identified concentrations of ~130–260 mM as optimal for later reactions.

With these conditions in hand, we scaled low mg reactions to a scale of 100 s of mg (from 100 mg [$^{19}$F]FDG). Reaction progress through each step in a one-pot operation was monitored by the measurement of the crude $^{19}$F-NMR spectra from reaction mixtures (Fig. 2C). This in turn allowed the testing of repeated batches (Fig. 2D) using fresh enzyme (including from newly expressed preparations) and reactants; we reasoned that such a batch scale would prove sufficient for a dose in man (see below for further details) and was therefore a representative test batch. These experiments revealed that at higher scale steps one and three (Fig. 2A) proved slightly more efficient than step two with the occasional observation of small amounts of unconverted deoxy-fluoro-G6P ([$^{19}$F]-**1**) in some batches (Fig. 2D) – advantageously, by virtue of its charge, if present [$^{19}$F]-**1** (and similarly [$^{18}$F]-**1**, see below) could be readily removed both during manual and automated syntheses (see below), further confirming the advantage of Route C. In this way we could readily and consistently access [$^{19}$F]FDT at 100 mg scales through one-pot biocatalytic syntheses; allowing the ready synthesis of grams of [$^{19}$F]FDT that when characterized (including via $^1$H-, $^{13}$C-, $^{19}$F-, $^{23}$Na- NMR, $^1$H qNMR, microanalysis and LC detected by multiple methods – see Figs. S16–29) complied fully with stringent measures of purity ( > 98% organic, with NaCl as the only co-eluent ( ~ 10–12%)) for subsequent in vivo toxicity testing (see below and Supplementary Methods).

Next, with 'cold' syntheses in hand, we applied essentially analogous methods for the one-pot syntheses of [$^{18}$F]FDT: first manually and then in an automated fashion that might prove suitable for good manufacturing practice (GMP) synthesis in Phase 1 studies. Using commercially available aqueous solutions of [$^{18}$F]FDG with activity of 5–15 mCi we observed full conversion to [$^{18}$F]FDT in 60 min. Purification was effected simply by adding ethanol to precipitate protein, filtration (5 μm filter) and then purification of the resulting filtrate using ion-exchange ('Luna' NH$_2$, see Supplementary Methods). Product [$^{18}$F]FDT was eluted with ethanol in a non-decay-corrected RCY over three steps of 34 ± 14 % (n = 5) and with a radiochemical purity >99% (Figs. 30) and an overall reaction time of 60 min.

To test possible, 'real world' application, these radiosyntheses were repeated from different batches of starting [$^{18}$F]FDG and also used directly in in vivo studies (see below). Depending on starting material concentration, activity and condition, overall production times for [$^{18}$F]FDT of 30–120 min were tested. For example, reaction times could be reliably reduced from 60 min to 30 min by increasing concentration of enzymes, (OtsA$^{Sf}$, OtsB$^{Sf}$ and hexokinase to 9, 15 and 45 μM, respectively) to ensure >99% conversion to provide comparable quantities and the same high purities of [$^{18}$F]FDT. No detectable, residual protein from the reaction was observed in the final product, as tested by Western blot ( < 1 μg/reaction, Figs. 31).

Next, after this successful manual standardization and variation, a fully automated synthesis was performed in a GE Tracerlab FX-N module (Fig. S32). Even higher, overall decay-corrected radiochemical yield (Table S2) was observed with higher amounts of starting [$^{18}$F]FDG; under typical conditions (100 mCi [$^{18}$F]FDG), 41 ± 4 % (n = 2) non-decay corrected yield of [$^{18}$F]FDT was obtained in 50 min. The identity of product [$^{18}$F]FDT was confirmed via co-injection of an authentic standard using LC-MS; radiochemical purity was >98%, determined by HPLC (Figs. S33, S34). Specific activity, assessed using an adapted enzymatic assay[47] of decayed samples (see Methods), revealed that, as expected, activity was essentially dependent on [$^{18}$F]FDG source – useful activities could be routinely achieved (average 0.21 ± 0.08 μmol [$^{18}$F]FDT at specific activity of 69 ± 26 mCi/mg (23.6 ± 8.8 Ci/mmol) at the end of synthesis and formulation; these varied from 8.3 – 86.8 Ci/mmol depending on the FDG batch (see Supplementary Methods).

## [$^{19}$F]FDT and [$^{18}$F]FDT display useful metabolic stability

For evaluations in vivo, solutions of [$^{19}$F]FDT were injected into both marmosets (n = 3) and mice (n = 5), and plasma subsequently analyzed (see Methods); both LC-MS and $^{19}$F-NMR spectroscopy allowed detection of [$^{19}$F]FDT and any putative degradation (Figs. S36, S37). Together, these suggested useful stability of [$^{19}$F]FDT both in plasma and metabolically in vivo with no detectable degradation. This suggested stability was confirmed for [$^{18}$F]FDT through direct assessment of radiochemical purity maintained >95% in both blood and urine of rabbits (Fig. S3E) (see also above and below). As noted above, this contrasted strongly with the significant metabolism of both 6-[$^{18}$F]FDT and 4-*epi*-[$^{18}$F]FDT.

Finally, in vitro [$^{19}$F]FDT in human plasma showed no immediate degradation or apparent sequestration and could also be readily detected (Fig. S35). Promising stability was observed ( > 50% after 80 min) in human liver microsomal degradation assays (Fig. S38).

## FDT labels mycolate lipids in Mtb

Although we have previously shown that FITC-Tre (Fig. 1B) can be incorporated into the cell surface of *Mtb* in cultured macrophages[21], its use for non-invasive in vivo imaging is not possible. We generated a $^{14}$C-radiotracer variant of [$^{19}$F]FDT, [U-$^{14}$C, $^{19}$F]FDT, using the chemoenzymatic methods detailed above, to allow observations over longer timeframes in *Mtb* culture. These Mtb-culture experiments revealed incorporation into less polar, lipidic species with TLC $R_f$ values that again match those of prior analyzes for similar mycolates[21,48], consistent with incorporation of FDT into corresponding [U-$^{14}$C, $^{19}$F]FDT-MM and [U-$^{14}$C, $^{19}$F]FDT-DM mycolates (Fig. S3F). Together these data suggested that FDT-based probes could act as probes that label mycobacterial lipids, in a manner consistent with the Mycobacterium-selective mechanism outlined in Fig. 1A.

## [$^{18}$F]FDT is a specific, consistent Mtb-radiotracer in vivo with uptake that correlates with bacterial burden in a non-human primate TB model

A useful PET probe for diagnosing tuberculosis should have minimal signal in the absence of *Mtb* infection, particularly in the lung where the majority of infections occur. Importantly, in a non-human primate (NHP) TB model[49] [$^{18}$F]FDT displayed low background signal in the lung of a naïve marmoset compared to ~45-day *Mtb*-infected marmoset (Fig. 3A–C). To confirm that this enhanced signal was specific to uptake of tracer, rather than, for example, non-specific uptake (proportional to tissue density) or normal distribution, 'pre-blocking' experiments were conducted with 'cold' [$^{19}$F]FDT (administered 1 h and 5 min) prior to injection of [$^{18}$F]FDT radiotracer. Consistent with specificity and mode-of-action, these reduced average uptake of [$^{18}$F]FDT into lesions by 40% (Fig. 3G, I, J). Since non-specific uptake resulting from transit or non-bound normal accumulation of the radiotracer should not be displaced by competition with blocker / 'cold' compound, a reduction in PET signal is consistent with specific binding.

Optimal imaging time was determined to be 90 min post-dose (Fig. 3); there was no additional improvement in signal-to-noise at 120 min scan. Reproducibility of [$^{18}$F]FDT uptake was assessed by measuring lesion uptake (SUV) in animals scanned 48 h apart (44 and

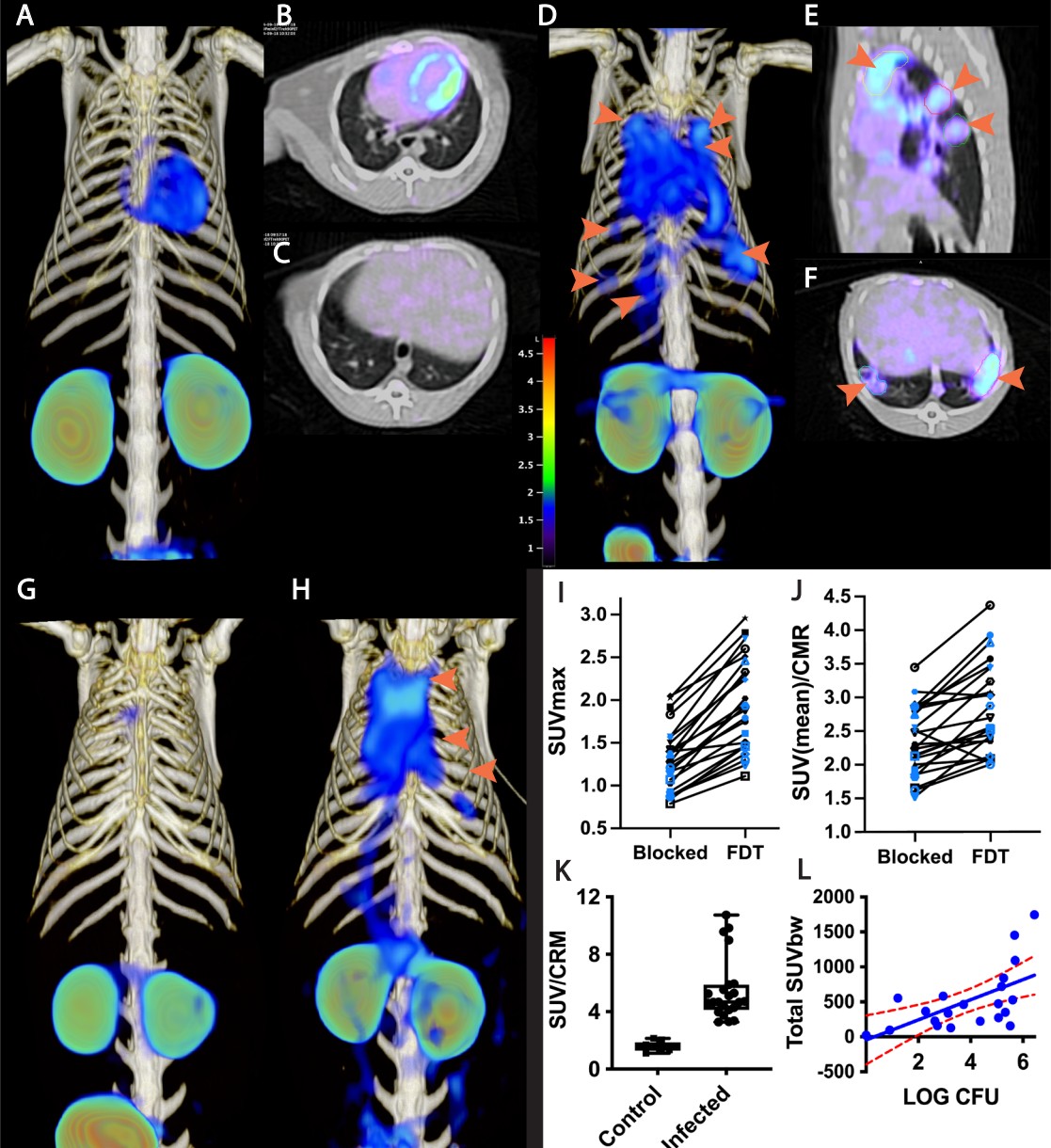

**Fig. 3 | Pulmonary [¹⁸F]FDT PET uptake into Tuberculous Lung Lesions is Reproducible within 90 min with Correlated Signal in Lesions with Higher Bacterial Loads. A–C** [¹⁸F]FDT PET/CT scan of a naïve marmoset lung (administered 1 mCi) [¹⁸F]FDT and imaged 90 min post-injection (3D volume rendering, transverse and sagittal views). **D–F** [¹⁸F]FDT PET/CT scan of a representative, *Mtb*-infected marmoset lung with lesions indicated by orange arrows (3D, transverse,and sagittal views). The target dose was 2.2 mCi/kg ( ~ 1 mCi). **G, H** 3D volume renderings of [¹⁸F]FDT PET uptake scans for blocked (**G**) and unblocked (**H**) uptake in the same infected marmoset. Marmosets (n = 2) were randomized as to the order of blocked scan; two days later the same animals were imaged with the treatments reversed. [¹⁸F]FDT uptake was blocked by administering excess [¹⁹F]FDT, split into two administrations 1 h and 5 min prior to radiotracer administration. **I, J** Reduction of [¹⁸F]FDT SUVmax and SUVmean/CMR, respectively, in marmosets administered cold [¹⁹F]FDT blocker prior to being administered [¹⁸F]FDT (1.2 mCi) compared to when administered [¹⁸F]FDT alone. In both cases the animal receiving [¹⁹F]FDT blocking dose showed significantly lower accumulation of [¹⁸F]FDT into tubercular

lesions (p < 0.0001, 2-tailed paired T for both measures; n = 24 lesions [12 individual lesions measured in each of the 2 animals scanned]). Different symbols have been used, colored black or blue, with one symbol-color combination for each animal. **K** The [¹⁸F]FDT PET signal from *Mtb*-infected marmoset lungs is significantly higher than the signal from uninfected (control) marmoset lungs. The mean Standard uptake values SUV/CRM are represented as boxplots (box bounds: lower and upper quartile range) where the central bars represent the median and whiskers show all points max to minimum. At least 4 lung regions of interest (ROI) were measured from 3 infected and 2 uninfected marmosets (p < 0.0001, two-tailed unpaired, T test [n = 22 ROIs in infected lungs; n = 9 in ROIs uninfected lungs]). **L** [¹⁸F]FDT uptake into tubercular lesion tends to increase with higher mycobacterial loads (p = 0.0019, Pearson's ρ = 0.63). The total SUV of each lesion ROI was compared with mycobacterial colony forming units (log CFU) measured in the 22 lesions (dots) collected from two marmosets euthanized within 24 h of FDT scan collection. The mean and 95% confidence interval lines (red) are shown on a linear regression plot. Please also see Source Data File.

46 days-post-infection (dpi)), sufficient time to allow full tracer clearance but short enough to minimize possible infection progression (Fig. 3); consistent [¹⁸F]FDT uptake (SUVs) was observed confirming reproducible uptake.

Finally, to determine whether lesion uptake of [¹⁸F]FDT correlated with disease, we determined bacterial burden at sites that were imaged by [¹⁸F]FDT. Post-necropsy excision of lesions (*n* = 22) revealed that [¹⁸F]FDT uptake into individual lesions (SUV per lesion) was

significantly correlated (p = 0.001, ρ = 0.64) with the number of culturable Mtb bacteria (CFU) from each lesion (Fig. 3K, L).

## [18F]FDT uptake and radiotracer signal differs from that of [18F]FDG in Mtb-infected models

We reasoned that this designed, mode-of-action of [18F]FDT that correlated well with mycobacterial burden might complement the differing mode-of-action shown by [18F]FDG uptake and hence reveal additional, valuable physiological information when used in tandem. In particular, [18F]FDG in TB appears to label immune cells[8,9] that are highly metabolically-active.

To the test the potential for different labeling patterns, we directly compared [18F]FDT and [18F]FDG scans collected sequentially in Mtb-infected marmosets (n = 4), with sufficient intervening time to clear (Fig. 4A–D). These revealed subtle but clear differences in the pattern of labeling in individual lesions and pooling in tissue regions with different characteristics. Fig. 4A shows an example where the apical and lower lesions labeled by [18F]FDG are clearly non-overlapping with those that label most strongly with [18F]FDT. In other cases (see Fig. 4B, C) where there were fewer lesions, these appeared to be similarly labeled with both probes, although FDG was consistently stronger (note scale bar is an MIP of 6 for [18F]FDG and 3 for [18F]FDT). In Fig. 4D, although there are only 5 days between the scans the disease has progressed significantly, and new lesions appear to label most intensely with [18F]FDT, suggesting that these represent areas of active bacterial replication. Further studies will be required to explore potential links between pathology and bacterial burden; these animals were instead used for exploring treatment response and meaningful necropsy samples were therefore unavailable.

## [18F]FDT allows direct monitoring of treatment that correlates with reduction in bacterial load

One of the most significant challenges facing current TB diagnosis and treatment programs is a source of reliable information on the disease burden in patients[50]. This is exacerbated by the often-prolonged treatment regimens that may continue over months and even years. Compliance with continual treatment is vital and direct feedback on progress of treatment would prove invaluable not only to the clinician but also patient[51,52]. Moreover, direct measurement of the in vivo effectiveness of new therapeutics would provide a powerful new tool in their clinical development[53–55].

We therefore tested whether [18F]FDT would report effectively on response to a representative Mtb therapy. Marmosets were treated for 4 weeks (five days per week) with first-line HRZE (combined isoniazid (H), rifampicin (R), pyrazinamide (Z) and ethambutol (E)) therapy and serial images were taken using both [18F]FDT (Fig. 5H–L) along with [18F]FDG as a comparator (Fig. 5A–E). Notably, [18F]FDG showed mixed, refractory and inconsistent uptake as judged by both SUVs and total lesion glycolysis (TLG) (Fig. 5F, G), despite the fact that effective treatment was confirmed by low bacterial burden after treatment (post-necropsy logCFUs of 2.6 and 2.8, respectively). By striking contrast, [18F]FDT scans showed a significant reduction (mean 33 ± 13% SD, p < 0.01) in SUV signal and a lower total [18F]FDT uptake (Fig. 5M, N) during treatment consistent with the determined drop in disease burden. Together, these data suggest that [18F]FDT could provide a more accurate probe of disease treatment.

## [18F]FDT is also an Effective Mtb-Radiotracer in an 'Old World' Non-Human Primate TB Model

Whilst, NHPs represent the most realistic pre-clinical TB models, species-specific differences are known[56]. To confirm the results (see above) found in the 'New World' (marmoset) NHP models, we also tested the [18F]FDT in 'Old World' cynomolgus macaques (n = 3) infected with Mtb (strain Erdman) (Fig. 6). TB in cynomolgus macaques is characterized by heterogeneous granulomas with a spectrum of Mtb burden, from lesions with many culturable bacilli to sterile lesions without live Mtb[57]. As in marmosets, [18F]FDT proved to be just as effective in specifically visualizing lesions (Fig. 6A) in a manner that

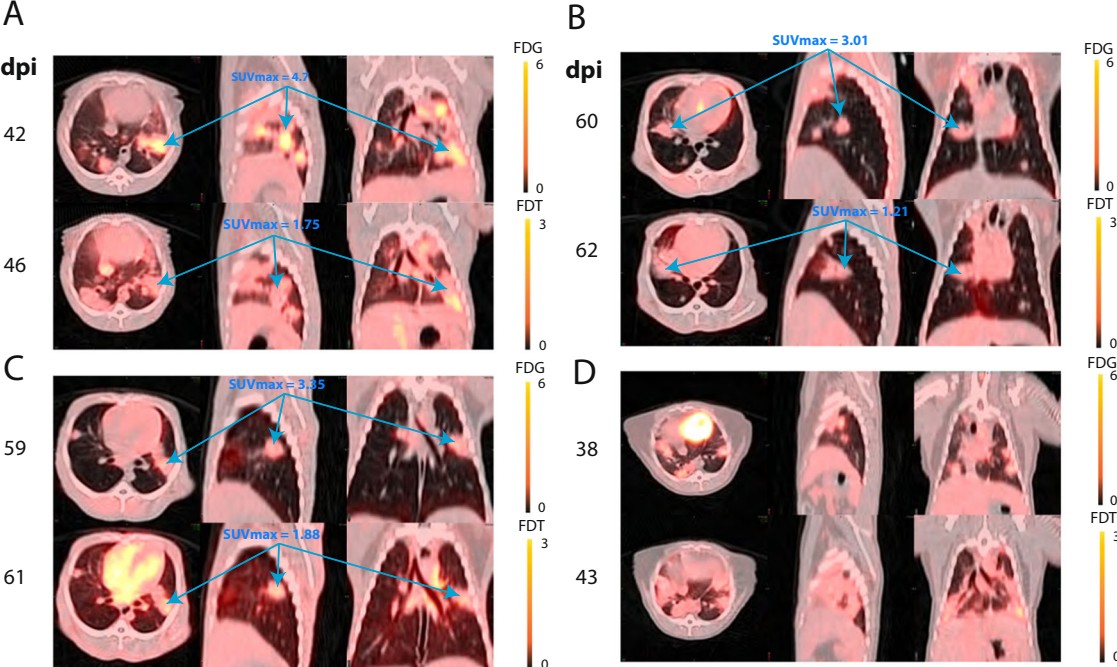

**Fig. 4 | Differential Uptake and Labeling of Tubercular Lesions by [18F]FDT and [18F]FDG in Mtb-Infected Marmosets.** Serial [18F]FDG (0.8 mCi) and [18F]FDT (1.1 mCi) scans of four TB-infected marmosets were collected at 90 min post injection **A**–**D**. Animals were imaged as close in time as possible allowing for decay of FDG and for the animal to recover from anesthesia. Fused images are shown with maximum intensity for [18F]FDG set to 6 and for [18F]FDT set to 3. SUVmax for some selected lesions for each probe are shown; [18F]FDT labeling intensity was typically about half that of [18F]FDG. **A** and **D**, despite being only four and five days apart show evidence of disease progression with newly evolved lesion areas appearing to label more intensely with [18F]FDT.

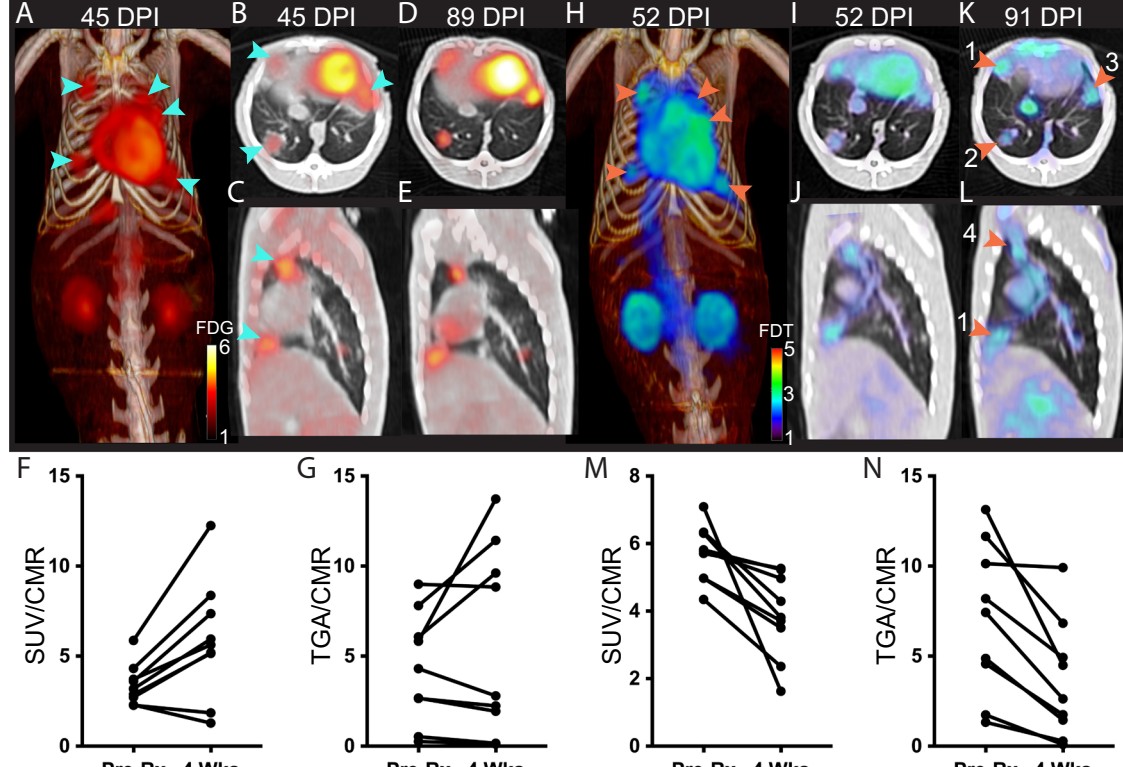

**Fig. 5 | [¹⁸F]FDT Scans of an *Mtb*-infected Marmoset Correlate with 4-week HRZE Treatment. A–G** 3D volume renderings and 2D axial and sagittal images and quantification of FDG uptake in Mtb-infected marmoset and the same marmoset given [¹⁸F]FDT scans with the number of days post infection (DPI) indicated above. **A–C**) In the pretreatment scan with [¹⁸F]FDG PET (0.7 mCi) collected 60 min post-injection, cyan arrows point to 5 lesions. **D–E** Marmoset imaged with [¹⁸F]FDG after 4 weeks of treatment showing the same 5 lesions. **F** [¹⁸F]FDG SUV before and after treatment of the lung lesions was variable and sometimes increased with treatment (p = 0.011, two-tailed paired T test, n = 9 lesions). **G** Total probe uptake as determined by total glycolytic activity (TGA) of the lesions showed a lesion-dependent pattern suggestive of avid [¹⁸F]FDG uptake, despite HRZE treatment (p = 0.239, two-tailed paired T test). **H–N** By contrast, similarly-timed (pretreatment and 4 weeks of treatment) [¹⁸F]FDT scans 90 min post injection (**I** 0.7 mCi; **K** 1 mCi) showed reduced SUV and total [¹⁸F]FDT uptake in the lesions after 4 weeks of treatment. Typical initial lesion bacterial load is 4 to 6 log CFU. Lower bacterial load was indeed observed in the lesions: lesion 1 = 2.3, lesion 2 = 2.6, lesion 3 = 3.1, lesion 4 = 0.9 log CFU and CFU is associated with FDT total uptake. **M** After HRZE combination drug therapy, the [¹⁸F]FDT SUV/CMR of the lung lesions was significantly decreased (p = 0.0069, two-tailed paired T test, *n* = 9 lesions) and **N** the total [¹⁸F]FDT uptake in the lesions, as determined by TGA, was also reduced (p = 0.0036, two-tailed paired T test). One representative image pair is shown of three similar treatment experiments. Please also see Source Data File.

again is distinct from and complements [¹⁸F]FDG (Fig. 6B). The uptake of [¹⁸F]FDT into cynomolgus macaques correlated well with bacterial disease burden, as lesions with culturable *Mtb* bacilli exhibited higher uptake of [¹⁸F]FDT (Fig. 6C, D).

## [¹⁹F]FDT is non-toxic and [¹⁸F]FDT shows rapid clearance resulting in low background dose exposure

No signs of toxicity or adverse effects were observed in any of the imaging studies described above. To confirm this apparent compatibility as well as determine the broader physiological characteristics of this novel radiotracer, both the biodistribution and toxicity of FDT were assessed in healthy animals.

First, dynamic PET imaging was performed in a healthy rhesus macaque for approximately 115 min after bolus intravenous administration of 151 MBq of [¹⁸F]FDT (Fig. 7A). This indicated rapid distribution of [¹⁸F]FDT from the blood stream through the tissues and prominent accumulation of radioactivity in kidney and urinary bladder, indicating renal excretion with minimal accumulation in other organs. Organ radiation absorbed doses were calculated by extrapolating the rhesus macaque data to humans, with organs receiving the highest doses being the urinary bladder wall (0.143 mSv/mBq; 2 hr voiding interval), kidneys (0.119 mSv/MBq), and adrenals (0.022 mSv/mBq) (Fig. 7B). The effective dose was 0.0154 mSv/MBq. Semi-comparative doses for a 1.5 hr voiding window for FDG are 0.086,

0.020 and 0.013 mSv/MBq, respectively; thus, there is a ~2-to-5-fold difference in exposure. Next, additionally, we performed biodistribution studies in two *Mtb*-infected marmosets injected with 74 MBq/kg [¹⁸F]FDT. Small samples of tissue were collected at necropsy 120 min post-injection and the radioactivity was measured in a calibrated gamma counter. In both animals, there was little tissue uptake of the radiotracer in organs other than in kidney (Fig. 7C).

Finally, single (acute) and multiple (chronic) intravenous (iv) dose toxicity studies of [¹⁹F]FDT, enabled by ready synthesis (see above), were conducted in both rats and beagle dogs (males and females both) via a contract with SRI International (Menlo Park, CA). The animals were given either daily iv injections of [¹⁹F]FDT at 100 × the expected human dose for seven consecutive days or a single iv injection at either 100 × or 1000 × the expected human dose once (see study schema tables in the methods). Mortality and morbidity, clinical observations, body weights, food consumption, hematology, serum chemistry, and coagulation parameters (beagles only), organ weights, and gross pathology / histopathology were evaluated daily for 9 days (acute toxicity) or 21 days (recovery group). There were no adverse findings in any parameters measured in the studies that were outside the expected range of normal. The no observed adverse effect level (NOAEL) in Sprague Dawley rats was at least 13.2 mg/kg when given as a single IV injection or 1.32 mg/kg/day when given by daily IV administration for 7 consecutive days and the maximum tolerated dose (MTD) was not

reached. The NOAEL in beagle dogs was at least 4 mg/kg when given as a single IV injection or 0.4 mg/kg/day when given by daily IV administration for 7 consecutive days and the MTD was not determined. All animals survived until scheduled euthanasia (day 9 or day 21) except for one rat with tail lesions that were judged to not be test article related and that was euthanized earlier.

## Discussion

Our results reveal that readily-generated [18F]FDT appears to complement FDG as an effective tracer of TB, with better selectivity and significant correlation to mycobacterial burden in lesions. We suggest that this selectivity is likely a result of its mechanism-based mode-of-action (Fig. 1A) that unlike FDG co-opts specific TB-associated enzymes not present in host, allowing selective incorporation into mycobacterial lipids. Nonetheless, FDG is a known, effective probe of TB. Together these observations suggest that FDG may report on inflammation proximal to but not located directly in a bacterial lesion, whereas [18F]FDT appears to report directly on site of *Mtb* enzymatic action (and therefore on bacterial viability). The lack of direct co-location (see Fig. 4) suggests that some tubercular lesions may therefore be seemingly less metabolically active than inflammatory hotspots. In this context, it may be that these differences are speculatively indicative of the physiology of host response, including the highly active balance of pro- and anti-inflammatory mediators[58], perhaps differentially driven by different cell subpopulations with different metabolic activities and locations within a lesion.

In the context of clinical evaluation of therapeutic response to TB treatment it is noteworthy that FDG avidity is maintained in a large fraction of cured patients even at the end of therapy[59]. This suggests that inflammation persists even after bacterial viability is lost and so limits the use of FDG-PET in determining the appropriate duration of treatment. Moreover, bacterial viability in sputum is lost months before patients can be safely discontinued without experiencing disease relapse; this suggests clear limits on the use of sputum sampling. Our proposed use of [18F]FDT-PET now offers the important potential to have real-time monitoring of bacterial viability in the absence of culturable bacteria and may be an important tool in creating novel treatment regimens of shorter duration.

Our results also suggest that use of [18F]FDT can now avoid potentially refractory sources of signal from non-TB-associated inflammatory responses that potentiate metabolism in a lesion-like region. This has long been a consideration in patients with other diseases associated with inflammation of the lung, for example cancer[11–13,60], but has been further highlighted by confusion generated from FDG-PET in putative COVID-19 patients with lung pathology[14]. Moreover, other granulomatous disorders, where histopathological distinction from TB proves difficult, such as Crohn's disease[61], might also now be more accurately diagnosed.

Together therefore our results suggest that disease-selectivity demonstrated by [18F]FDT will allow effective monitoring of disease and its treatment. This is likely to be invaluable for clinical evaluation of not only effectiveness in the development of new therapies but also in longitudinal monitoring and hence compliance. The limitations of our study include the use of only small numbers of primates due to reduced number of scans that can be conducted with animals bearing untreated progressive tuberculosis. In addition, the efficacy of [18F]FDT in the presence of other possible lung diseases has yet to be assessed; although direct comparison with [18F]FDT should be made with caution given potentially differential processing, we have previously determined that the analog FITC-Tre does not label several other bacterial species that cause lung infections[21]. It should be noted that such studies will be required in Phase 1 clinical studies. In this context the additional roles of other trehalose pathways, such as the *lpqY-sugA-sugB-sugC*-derived ABC transporter system may further modulate selectivity[37,62].

The scaleable, pyrogen-free radiosynthesis methods that we describe here now confirm a proposed use[21] of highly selective biocatalysis under aqueous conditions that are readily implemented by the non-expert. Other elegant methods[37,38,63] and newly emerging enzyme sources[64] could also be considered. Such biocatalytic approaches to utilization of FDG as a ready organic source of 18F now complements chemical methods, such as the use of 18F-fluorodeoxysorbitol (FDS) that can be directly generated in a one-step reduction and has shown powerful utility in other bacterial species such as *E. coli*[65]; the extension here to a ready, one-pot multi-step method now reveals the potential to consider even more complex sugar-based probes. This now enables potential, distributable radiochemical synthesis of [18F]FDT to be conducted anywhere there is access to FDG; scales of up to grams have now been demonstrated using this method for [19F]FDT. Furthermore, full, pre-clinical assessments (see above) reveal no adverse effects. This, as well as the high specific activities and good radiochemical efficiencies that we disclose here now suggest [18F]FDT as a new, viable radiotracer for TB, suitable for Phase 1 trials.

## Methods
### Materials

Non-radioactive [19F]FDG was purchased from CarboSynth. For radioactive synthesis, [18F]FDG was purchased from Cardinal Health Ltd. Normal saline was obtained from Quality Biological (Gaithersburg, MD, USA). All other chemicals and solvents were received from Sigma-Aldrich (St. Louis, MO, USA) and used without further purification. Column and Sep-Pak cartridges used in this synthesis were obtained from Agilent Technologies (Santa Clara, CA, USA) and Waters (Milford, MA, USA), respectively. Sep-Paks were conditioned prior to use with 5 mL absolute ethanol. Analytical HPLC analyses for radiochemical work were performed on an Agilent 1200 Series instrument equipped with multi-wavelength detectors. Mass spectra (MS) of decayed [18F]-FDT solutions were recorded on a 6130 Quadrupole LC/MS, Agilent Technologies instrument equipped with a diode array detector. LC-MS analysis of [18F]-FDT was performed on Agilent 1260 HPLC system coupled to an Advion expression LCMS mass spectrometer with an ESI source. The LC inlet was Agilent 1200 series chromatographic system equipped with 1260 quaternary pump, 1260 Infinity autosampler, 1290 thermostatted column compartment and radiation detector. Further details of chromatographic conditions are given in Tables S3–S5. Hexokinase enzyme was purchased from Megazyme. Detailed procedures for expression and purification of other enzymes (OtsA also known as TPS, OtsB also known as TPP, OtsAB fusion and TreT variants) both in *E. coli* and Sf9 insect cells (the latter denoted by use of *Sf*) are given in the Supplementary Methods and below.

### Enzyme Kinetics

Kinetic parameters of TreT-catalyzed synthesis of trehalose were assayed continuously by coupling the formation of UDP to the reactions of pyruvate kinase and lactate dehydrogenase. The decrease in absorbance of NADH at 340 nm ($\epsilon 340 = 6220\ M^{-1}cm^{-1}$) was measured at 37 °C using a spectrophotometer equipped with a thermostat. The rate of UDP formation is proportional to the rate of NADH oxidation where one molecule of NADH is oxidized for each molecule of UDP formed. As the reaction involves two substrates, therefore, the kinetic parameters determined based on pseudo substrate concentration. TreT enzymes were incubated in the presence of fixed acceptor concentration and variable donor concentration, followed by fixed donor concentration and variable acceptor concentration. For continuous assay, each reaction was measured as triplicates by setting up the reactions in 96 well plates. Reactions were initiated by adding TreT enzyme and incubated at 37 °C (see Table S6 and Supplementary Methods for further details).

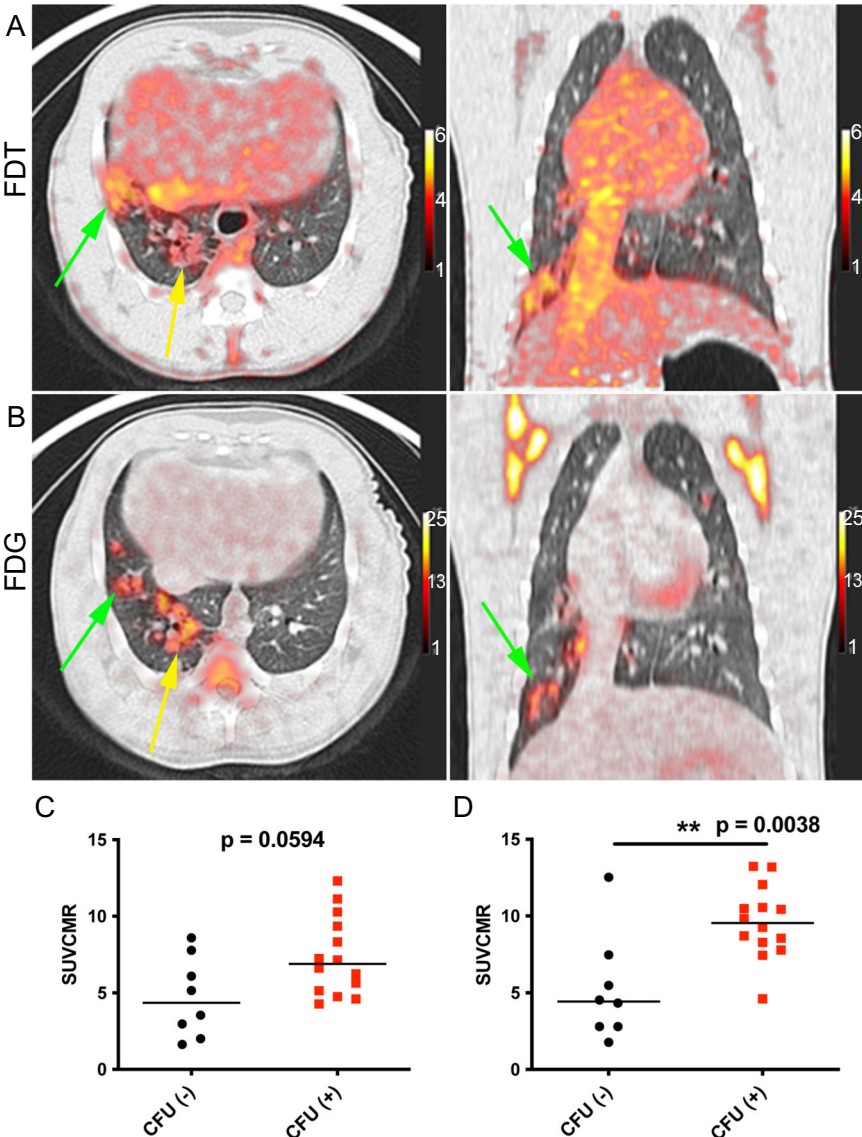

**Fig. 6 | [¹⁸F]FDT is also an Effective and Specific Radiotracer in an 'Old World' NHP TB Model. A** Axial and coronal views of the lung of a representative Mtb-infected cynomolgus macaque showing labeling of tubercular lesion clusters (green and yellow arrows) with [¹⁸F]FDT (5 mCi) scanned 60 min post-injection. Calibration scale in SUV. **B** Axial and coronal views of the same lung lesion clusters labeled with [¹⁸F]FDG (5 mCi) 60 min post-injection. Separate calibration scales in SUV. **C** The animal was necropsied and the individual lesions were plated. *Ante mortum* probe uptake of the lesions was compared to their culture status [colony-forming units negative CFU(-), black circles or positive CFU(+), red squares]. In images captured at 60 min there was a trend toward lesions with culturable bacteria having higher probe uptake. **D** By 120 min, the uptake of [¹⁸F]FDT was significantly higher among lesions with culturable mycobacteria than among sterile lesions. Quantitative data from two macaques (22 lesions total) imaged are shown with p values from two-tailed Mann-Whitney test. Imaging data from one of three representative animals are shown. Please also see Source Data File.

OtsA kinetic parameters were also determined by monitoring the formation of T6P by NADH-linked continuous assay. The rate of T6P formation is proportional to the rate of NADH oxidation where one molecule of NADH is oxidized for each molecule of T6P formed. OtsB-catalyzed trehalose formation was monitored by Malachite-green assay protocol as described previously[66]. Some further details of parameters used for optimization of the methods are given in Tables S7, S8).

For the NADH-linked based assays, the absorbance values obtained from the linear range of concentrations were plotted against time to determine the rate of the reaction (velocity). Plots of variable substrate concentrations against rate were analyzed in Graphpad Prism to obtain Michaelis-Menten plots. From these plots kinetic parameters of each enzyme were calculated. All results shown are based on triplicate analysis of the samples.

### Enzyme production scale-up in Sf9 cells expression

In order to obtain a reasonable expression level of OtsA and OtsB proteins, the Bac-to-Bac® Baculovirus Expression System was selected. The original coding sequence of *E. coli* OtsA (*otsA*, CAA48913.1) and OtsB (*otsB*, CAA48912.1) was optimized according to *S. frugiperda* MNPV codon usage. The DNA fragment was then cloned into the *EcoRV* restriction enzyme site of the pUC57 vector. Then the plasmids pUC57-OtsA and pUC57-OtsB and pFastBac Dual vector were digested with *BamHI* and *NotI*, respectively, and ligated to create the recombinant plasmids of OtsA (pFastBac-OtsA) and OtsB (pFastBac-OtsB). The bacmid DNAs of OtsA (Bacmid-OtsA) and OtsB (Bacmid-OtsB) were generated by transformation of the recombinant plasmids pFastBac-OtsA and pFastBac-OtsB into the DH10Bac™ *E. coli*, respectively. Correct insertions of the *otsA* and *otsB* genes into bacmid DNA were confirmed by PCR analysis using pUC/

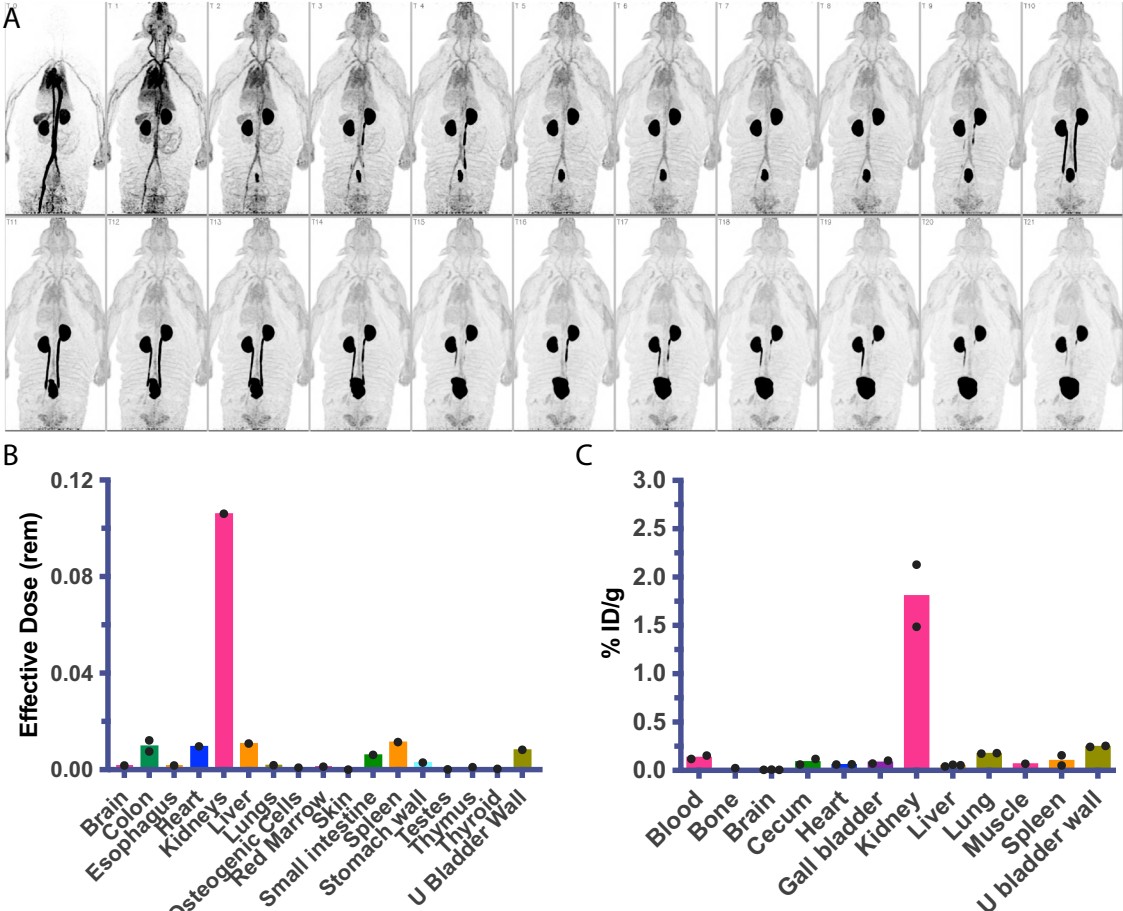

**Fig. 7 | Dynamic Biodistribution of [¹⁸F]FDT in Naïve Rhesus Macaques and Infected Marmosets to Estimate Organ Exposure. A** Sequential coronal maximum intensity projections of [¹⁸F]FDT PET activity in a representative naïve rhesus macaque collected over approximately 115 min in scan frames of increasing duration in a dynamic PET scan (from top left) after administration of 151 MBq of [¹⁸F]FDT. **B** Organ radiation absorbed doses (mSv/MBq) calculated by extrapolating the rhesus macaque data ($n = 1$) to humans; organs receiving the highest doses were

urinary bladder wall, kidneys, and adrenals. **C** Similar to the rhesus experiments, the quantification of the radioactivity in marmoset tissues (using gamma counting of [¹⁸F]FDT radioactivity in excised tissues of euthanised marmosets, $n = 2$) indicates the kidneys were the solid organs with the highest proportion of the injected dose (%ID/g) when marmosets were necropsied 120 min after being administered [¹⁸F] FDT. Please also see Source Data File.

M13 standard primers: 5-CCCAGTCACGACGTTGTAAAACG-3 (forward) and 5-AGCGGATAACAATTTCACACAGG-3 (reverse).

Based on the bacmid DNAs, the recombinant baculovirus of OtsA and OtsB were made by transfection of Sf9 cells. Culture and preparation of Sf9 cells were performed according to the protocol provided by manufacturer. A Sf9 cell stock with a viability >95% and a density of $1.5 \times 10^{6} - 2.5 \times 10^{6}$ cells/mL was prepared before proceeding to transfection.

### Experimental procedures to generate P1 baculovirus for OtsA and OtsB

The P1 baculovirus stocks of OtsA and OtsB were generated using Sf9 cells in 6-well culture plates. Briefly, 2 ml of Sf9 cells were added to each well of two 6-well plates with the cell number of $8 \times 10^{5}$ in each well. Cells were allowed to attach for 15 min at room temperature. Transfectin II was used as the transfection reagent and the transfection experiments were performed according to the protocol provided by manufacturer. Different amount of bacmid DNAs of OtsA and OtsB were used. After transfection, the Sf9 cells were further cultured at 27 °C for 3 days, then the P1 baculovirus (in supernatant) was harvested by centrifugation. The P1 baculovirus are found in the culture medium and the Sf9 cell pellets were used to check protein expression. Fetal bovine serum (FBS) was then added to each P1 baculovirus solution to

reach a final concentration of 2%. The P1 baculovirus stock solution were then stored at 4 °C, protected from light.

Amplification of P1 baculovirus of OtsA and OtsB: In order to make baculovirus stock for large-scale protein expression and to increase virus titer, P2 and P3 baculovrius stocks of OtsA and OtsB were prepared from their P1 baculovirus. Briefly, a total of 150 mL of suspended Sf9 cells with a cell density of $2.0 \times 10^{6}$ cells/mL was infected by P1 baculovirus of OtsA or OtsB, respectively. The 150 mL infection culture composition was: 1% P1 baculovirus of OtsA or OtsB, 1% FBS, 1% ethanol, Sf9 cells with a density of $2.0 \times 10^{6}$ cells/mL. After infection at 27 °C with shaking (150 rpm) for 72 h, the P2 baculovirus (in culture medium) was harvested by centrifugation at 1100 rpm for 10 min at room temperature. FBS was added to a final concentration of 2%, then the solution was filtered by syringe filter (0.45 μm) and stored at 4 °C, protected from light. Similar infection procedures were performed to prepare P3 baculovirus, except using P2 baculovirus stock solution for cell infection.

### Expression and Purification of C-terminally tagged, recombinant OtsA

Sf9 cells were grown in serum-free medium to a density of $2 \times 10^{6}$ cells/mL and a cell viability of greater than 95%. The cell culture (1 L) was transfected with the viral stock, $2.8 \times 10^{9}$ pfu/mL, at a multiplicity of

infection (MOI) of unity. At 72 h post-infection, the cells were harvested by centrifugation. After freeze/thaw cycle, the pellet was resuspended in lysis buffer (100 mM HEPES, 200 mM NaCl, 10 mM imidazole, 10 mM MgCl$_2$, 1 mM βME). DNase 3 mg and 1x Complete® protease inhibitor cocktail was also added into the protein pellet and lysed using a homogenizer. Cell debris was removed by centrifugation and the cell free extract passed through a 0.45 μm membrane filter. The filtrate was applied to a Ni−NTA column equilibrated in lysis buffer. After removal of the cell debris by centrifugation, the protein of interest was purified with a linear gradient of 10−300 mM imidazole in 100 mM HEPES, 200 mM NaCl, 10 mM MgCl$_2$, 1 mM βME, pH 7.5, using Ni-NTA. Protein samples were exchanged into 100 mM Hepes, 150 mM NaCl, 10 mM MgCl$_2$, pH 7.5, using a Hiload™ 26/60 Superdex™ 200 desalting column. The sample was passed through an Acrodisc Mustang E membrane (Pall #MSTG25E3) to reduce endotoxin to 1.9 EU/mL. After the addition of 10% glycerol, the aliquots containing the recombinant enzyme were flash frozen, and stored at −80 °C.

## Expression and purification of recombinant OtsB

Insect cells were grown in serum-free medium to a density of $2 \times 10^6$ cells/mL and a cell viability of greater than 95%. The cell culture (1 L) was transfected with the viral stock, $2.0 \times 10^9$ pfu/mL, at a multiplicity of infection (MOI) of unity. At 72 h post-infection, the cells were harvested by centrifugation. After the freeze/thaw cycle, the pellet was resuspended in lysis buffer (50 mM Tris-HCl, 200 mM NaCl, 5 mM MgCl$_2$, 30 mM imidazole, 1 mM βME). DNase 3 mg, 1x Complete® protease inhibitor cocktail was also added in the pellet and lysed using a homogenizer. Cell debris was removed by centrifugation and the cell free extract passed through a 0.45 μm membrane filter. The filtrate was applied to a Ni−NTA column equilibrated in lysis buffer. After removal of the cell debris by centrifugation, the protein of interest was purified with a linear gradient of 10−300 mM imidazole in 50 mM Tris-HCl, 200 mM NaCl, 5 mM MgCl$_2$, 1 mM βME, pH 7.5, using Ni-NTA. The fractions with potential protein were concentrated using Vivaspin (5 MWCO) and then desalted on Hiload™ 26/60 Superdex™ 200 with 50 mM Tris-HCl, 5 mM MgCl$_2$, 100 mM NaCl, pH 7.5. The sample was passed through an Acrodisc Mustang E membrane (Pall #MSTG25E3) to reduce endotoxin to 1.9 EU/mL. After the addition of 10% glycerol, the aliquots containing the recombinant enzyme were flash frozen, and stored at −80 °C. SDS–PAGE analysis revealed the presence of a protein with the correct molecular weight.

## FDT Synthesis, purification and characterization

In a 50 mL sterile falcon tube were added; $^{19}$F-2-FDG (100 mg, 367 mM), ATP (320 mg, 368 mM) and MgCl$_2$ (2 mg). The contents were dissolved in HEPES (100 mM HEPES) buffer solution containing 100 mM NaCl (100 mM, 1 ml, pH 7.5). The pH was then adjusted to 8.0 with NaOH (5 M). Hexokinase (0.6 mL) was added to the reaction mixture to initiate the reaction. The reaction mixture was incubated at 30 °C, 50 rpm for 1–2 h. Phosphorylation of FDG to 2-FDG6P was monitored by TLC and $^{19}$F NMR spectroscopy. Full conversion to 2-FDG6P was observed within 2 h for each batch synthesized.

Within the same pot, UDP-glucose (200 mg, 110 mM), KCl (100 mM) and OtsA enzyme (26 μM) was added. The total reaction volume was then adjusted to 3 mL. The reaction mixture was incubated to similar conditions as above for 24 h to allow appropriate conversion to FDT6P. The conversion of FDG6P to FDT6P was monitored by $^{19}$F-NMR spectroscopy. The average maximum conversion after 24 h was between 65–70 %. After 24 h, in the same reaction mixture OtsB enzyme (6.23 μM) was added. The pH was adjusted to 8.0 with NaOH (5 M) and reaction mixture was incubated at 37 °C, 50 rpm. Reaction progress was monitored by TLC and $^{19}$F NMR. The reaction was usually complete within 2 h.

The enzymes from crude reaction mixture were precipitated by either heating the mixture at 95 °C for 10 min or adding 2 mL EtOH and

isolated by centrifugation (Viva spin, 10,000 MWCO) and/or filtration. The resulting filtrate was lyophilized and the crude mixture was further precipitated with MeOH:MeCN 90:10 solution and then purified by flash column chromatography. For purification normal phase silica was used as stationary phase and a mixture of water:isopropanol:ethyl acetate (4:48:48 v:v:v) was used as mobile phase. FDT was separated from residual FDG6P and residual FDG. The fractions containing FDT were pooled, concentrated and then lyophilized to dryness. The lyophilization step was repeated for a further 2–3 times to get rid of any residual water. The freeze-dried compound was then characterized by proton ($^1$H NMR), carbon ($^{13}$C NMR), fluorine ($^{19}$F NMR) and ESI-MS.

## Manual Radiosynthesis and characterization of [$^{18}$F]-FDT

For radiosynthesis using 3 enzymes, to a 1.5 or 2 mL microcentrifuge tube containing a solution of commercial [$^{18}$F]-FDG in 0.9 % NaCl, reactants were added sequentially to achieve final concentration of UDP-glucose 30 mM, MgCl$_2$ 10 mM, KCl 100 mM, ATP 10 mM, Hexokinase 640 U, OtsA 0.7 mg and OtsB 0.4 mg and enough HEPES buffer (100 mM HEPES, 150 mM NaCl and 10 mM MgCl$_2$, pH 7.5) to achieve the desired volume of 1 mL. After gentle mixing, the tube was capped and incubated for 60 min at 37 °C in a thermoshaker inside the hot cell. Reaction progress was monitored by radio-HPLC during this period. After 40 min, reaction was almost 80 % completed. After 60 min, the enzymes in the reaction mixtures were precipitated by adding ethanol solution, filtered through 5 μm filter and then purified by Luna-NH$_2$ Sep-Pak cartridge method (details in the Supplementary Methods).

Radiochemical yield of the purified product was calculated by assaying the activity (mCi) of the vial, reported as non-decay corrected yield and is based on starting activity of [$^{18}$F]-FDG. The radiochemical purity of the product was analyzed by radio-HPLC using an Agilent 1200 series HPLC coupled with radio detector for monitoring the radioactivity. Radiochemical purity was determined by calculating the area under the curves in the radiochromatogram.

For radiosynthesis using 1 enzyme i.e. TreT similar procedure as described above was adopted. TreT being thermophile allowed the incubation of the reaction at higher temperature, therefore we carried out the reactions both at 37 °C and 60 °C.

## Chemoenzymatic synthesis of [$^{18}$F]-FDT for the animal models

For the biodistribution and blocking studies, [$^{18}$F]-FDG (20–30 mCi in 0.8 −1 mL) was added to the reaction mixture containing 100 μL 1 M HEPES buffer, pH 7.6, 20 μL 1 M MgCl$_2$, 20 μL 1 M ATP, 60 μL 1 M UDP-glucose, -50 μL OtsA (1 mg), -50 μL OtsB (1 mg), 20 μL hexokinase (5 mg). The reaction mixture was incubated at 37 °C for 30 min. After 30 min, the mixture was diluted with absolute ethanol (4 mL) and passed through a 5 μm syringe filter. The eluent was passed slowly through an amine Sep-Pak SPE cartridge at a flow rate of 1–2 drops per second. The eluent was then concentrated in vacuo. The resulting solution was filter-sterilized into a sterile vial for delivery. Identity of the compound was confirmed by LC-MS.

## Automated Synthesis of [$^{18}$F]-FDT

[$^{18}$F]-FDG (1480- 3700 MBq; 40–100 mCi in 0.8 −1.8 mL) was transferred under vacuum to the Reactor 1 containing 100 μL 1 M HEPES buffer, pH 7.6, 20 μL 1 M MgCl$_2$, 20 μL 1 M ATP, 60 μL 1 M UDP-glucose, -50 μL OtsA (1 mg), -50 μL OtsB (1 mg), 20 μL hexokinase (5 mg). The reaction mixture was incubated for 30 min at 45 °C and absolute ethanol was added (3–6 mL). The reaction mixture was passed through the filter (5 μm) and a stack of three NH$_2$-cartridges and collected in Reactor 2. An additional 1 mL 75% ethanol in water was added to rinse the Reactor 1 and transferred into the Reactor 2. The combined solution was concentrated under nitrogen at 60 °C for 10 min. 2 mL saline was added to the Reactor 2. The final [$^{18}$F]-FDT solution was transferred to the product vial through a sterile filter (0.22 μm,). The quality of the

product was determined by analytical HPLC (Condition: 4.6 × 250 mm, 2.7 μm AdvanceBio Glycan Mapping Column; solvent A = 50 mM ammonium formate, pH 4.5, solvent B = acetonitrile; Flow rate = 0.5 mL/min;gradient 0–15 min 68-62% B; 15–20 min 62–68% B). Identity of the compound was confirmed by LC-MS (Fig. 32).

## Specific activity analysis of [$^{18}$F]FDT

The radioactivity of the final products, [$^{18}$F]-FDT, was obtained using the radiation detector. To quantify the mass of the decayed [$^{18}$F]-FDT in the form of [$^{18}$O]trehalose, a modified version of a trehalose quantification enzymatic assay[47] was used. A trehalose standard calibration curve (linear fit, $R^2$ = 0.9984) was generated, following the enzymatic assay using known concentrations of trehalose. The decayed masses of [$^{18}$F]FDT from the syntheses were calculated based on this calibration curve. Then, the specific activity of the final product was calculated, following the definition, radioactivity at the end of the synthesis/unit mass of compound.

## Chemical Synthesis of 4-O-Triflyl-hepta-O-acetyl-trehalose (4-OTf-TreAc₇)

In a 10 mL tube, 50 mg of 6,3,2,6′,4′,3′,2′-hepta-O-acetyl-trehalose (0.078 mmol, 1 eq) and 24 mg of 2,6-di-t-butyl-4-methylpyridine (0.018 mmol, 1.5 eq) were dissolved in 3 mL of anhydrous dichloromethane. The mixture was cooled to 0 °C and 17 μL of triflic anhydride (0.012 mmol, 1.3 eq) was added slowly. After stirring at 0 °C for 1 h, the mixture was warmed to room temperature for 3 d. The reaction mixture was evaporated to dryness at low temperature and purified by silica column chromatography (9:1 CH₂Cl₂:EtOAc and then 7:3 of the same mixture, elution completed 15 min after loading) to give 25.9 mg of 4-OTf-TreAc₇ as a white solid in 43% yield. TLC (1:1 CH₂Cl₂:EtOAc) TLC: Rf = 0.58 (1:1 CH₂Cl₂:EtOAc, starting material Rf = 0.40);¹H NMR (400 MHz CDCl₃) δ ppm 2.04, 2.07, 2.08, 2.08, 2.11, 2.13 (7x 3H, s, CH₃ OAc), 4.02 (1 H, dd, J6b′,6a′ = 12.0 Hz, J6b′,5′ 2.4 Hz, H-6b′ CH₂OAc), 4.06 (1H, ddd, J5′,4′ = 10.4 Hz, J5′,6a′ 5.6 Hz, J5′,6b′ = 2.4 Hz, H-5′), 4.22 (3 H, m, H-5, H-6b, H-6a), 4.99 (1H, dd, J4,5 10.0 Hz, J4,3 9.4 Hz, H-4) 5.03 (1 H, dd, J2,3 = 10.4 Hz, J2,1 4.0 Hz, H-2), 5.06 (1H, dd, J2′,3′ = 10.4 Hz, J2′,1′ 4.0 Hz, H-2′), 5.07 (1 H, dd, J4′,5′ = 10.4 Hz, J4′,3′ 9.6 Hz, H-4′), 5.27 (1H, d, J1,2 4.0 Hz, H-1), 5.29 (1H, d, J1′,2′ = 4.0 Hz, H-1′), 5.48 (1H, dd, J3′,2′ = 10.4 Hz, J3′,4′ 9.6 Hz, H-3′), 5.70 (1H, dd, J3,2 = 10.4 Hz, J3,2 9.4 Hz, H-3); ¹³C NMR (101 MHz CDCl₃) δ ppm 20.4, 20.5, 20.6, 20.6, 20.6, (7x CH₃ acetates), 61.0 (C-6), 61.6 (C-6′), 67.2 (C-5), 68.3, 68.3 (C-5′), 68.4, 68.4 (C-3, C-4′), 69.8 (C-2 and C-2′), 70.0 (C-3′), 78.7 (C-4 OTf), 92.3 (C-1), 92.8 (C-1′), 169.3, 169.5, 169.8, 170.1 (7x C = O acetates); ¹⁹F NMR (376.5 MHz, CD₃OD):−74.4, s, CF3 OTf; HRMS ESI⁺ [C₂₇H₃₅O₂₀F₃S + Na]⁺ requires 791.1287, found 791.1246.

## Chemical Synthesis of 6-O-Triflyl-hepta-O-acetyl-trehalose (6-OTf-TreAc₇)

In a 50 mL flask, 160.6 mg of 4,3,2,6′,4′,3′,2′-hepta-O-acetyl-trehalose (0.252 mmol, 1 eq) and 77 mg of 2,6-di-t-butyl-4-methylpyridine (0.378 mmol, 1.5 eq) were dissolved in 10 mL of anhydrous dichloromethane. The mixture was cooled to 0 °C and 55 μL of triflic anhydride (0.328 mmol, 1.3 eq) was added slowly. The mixture was stirred while monitoring by TLC, 1 h at 0 °C followed by 24 h at r.t. The reaction mixture was evaporated to dryness at low temperature and purified by silica column chromatography (9:1 CH₂Cl₂:EtOAc and then 7:3 of the same mixture, elution completed 15 min after loading) to give 93.3 mg of 6-OTf Tre Ac₇ as a white solid in 48% yield. TLC (1:1 CH₂Cl₂:EtOAc) Rf=0.58 for the product and Rf=0.40 for starting material 6-OH Tre Ac₇. ¹H NMR (400 MHz CDCl₃-d) δ ppm 2.03, 2.03, 2.04, 2.05, 2.08, 2.09, 2.09 (7x 3H, s, CH₃ OAc), 3.99 (1 H, dd, J6b′,6a′ = 12.0 Hz, J6b′,5′ 2.4 Hz, H-6b′ CH₂OAc), 4.05 (1H, ddd, J5′,4′ = 10 Hz, J5′,6a′ 5.6 Hz, J5′,6b′ = 2.4 Hz, H-5′) 4.22 (1H, m, H-5) 4.25 (1 H, m, H-6a′ CH₂OAc), 4.41 (1 H, dd, J6b,6a = 11.2 Hz, J6b,5 2.4 Hz, H-6b

CH₂OTf), 4.52 (1 H, dd, J6a,6b = 11.2 Hz, J6a,5 6 Hz, H-6a CH₂OTf), 5.00–5.10 (4H, m, H-2, H-2′, H-4, H-4′), 5.28 (1H, d, J1′,2′ 3.6 Hz, H-1′), 5.33 (1H, d, J1,2 3.6 Hz, H-1), 5.48 (1H, dd, J3,4 = 10.0 Hz, J3,2 9.6 Hz, H-3), 5.50 (1H, dd, J3′,4′ = 10.0 Hz, J3′,2′ 9.6 Hz, H-3′); ¹³C NMR (101 MHz CDCl₃-d) δ ppm 20.6, 20.7, 20.7 (7x CH₃ acetates), 61.7 (C-6′ OAc), 67.8 (C-5), 68.3 (C-5′), 68.4 (C-4), 68.4 (C-4′), 69.4 (C-2, C2′), 69.5 (C-3′), 70.0 (C-3), 73.3 (C-6 OTf), 92.4 (C-1), 93.1 (C-1′), 169.3, 169.4, 169.5, 169.6, 169.9, 170.0, 170.6 (7x C = O acetates); ¹⁹F NMR (376.5 MHz, CD₃OD):−74.4, s, CF₃ OTf; HRMS (ESI⁺) calcd. for C₂₇H₃₅F₃O₂₀SNa⁺ (M +Na⁺): 791.1287, found 791.1259

## Animal and ethics assurance

This study was carried out in accordance with the recommendations in the Guide for the Care and Use of Laboratory Animals of the National Institutes of Health. Biodistribution studies with naive rhesus macaques were approved by the Institutional Animal Care and Use Committee (IACUC) of the NIH Clinical Centre (Bethesda, MD). The IACUC of the NIAID, NIH approved the experiments described herein with rabbits and marmosets under protocols LCID-3 and LCID-9 respectively (Permit issued to NIH Intramural Research Program as A-4149-01), and all efforts were made to provide intellectual and physical enrichment and minimize suffering. Once infected, female rabbits or marmosets of both sexes were housed individually or paired in biocontainment cages in a biological level 3 animal facility approved for the containment of *M. tuberculosis*. All NIH studies were performed in accordance with the regulations of the Division of Radiation Safety, at the National Institutes of Health (Bethesda, MD, USA). Two uninfected and eight infected marmosets on protocol LCIM-9 were imaged with **[$^{18}$F]FDT** one or more times as well as contributing to other experiments. The marmosets were pair-housed in an approved ABSL3 facility at NIH. Three rhesus macaques from the NIH PET center were imaged dynamically with FDT under protocol PET-14-01. FDT studies in three cynomolgus macaque at University of Pittsburgh were approved by its IACUC and Division of Radiation Safety and all MTB infected animals were pair-housed in an approved ABSL3 facility (Regional Biocontainment Facility, Pittsburgh PA).

## Detection of serum [$^{19}$F]FDT

FDT standards were prepared in human plasma (Sigma Aldrich), whereby 20 μL plasma samples were spiked with various known concentrations of [$^{19}$F]FDT i.e. 10 μM, 20 μM, 40 μM, 120 μM, 160 μM, and 300 μM in 1.5 ml Eppendorf tube and diluted with up to 50% acetonitrile solution. Samples were vortexed for 1 min and centrifuged at 13000 rpm (15871 x *g*) for 10 min and supernatant was analyzed directly by LC-MS using LC method 2.

For initial in vivo study, six white mice, CD1, male, weighing between 22–26 grams, were used. Two concentrations of [$^{19}$F]FDT (500 μM and 50 μM, final concentration in the blood) were injected into the mice, through the tail vein injection. Prior to injection, each mouse was pre-bled (100 μL) and control samples were obtained. Five min post injection, blood samples were drawn up (775 μL) in syringe containing 75 μL sodium citrate as anticoagulant, hence total blood volume was 700 μL. Samples were spun at 13000 rpm (15871 x *g*) for 10 min and the supernatant (plasma) was used for further analysis. 200 μL of plasma sample was transferred into fresh 1.5 mL Eppendorf tube and diluted with equal amount of acetonitrile to precipitate out blood plasma protein and other macromolecules. Samples were further spun at 13000 rpm (15871 x *g*) for 5 min. Supernatant was transferred into a mass spectrometry vial and analyzed using LC-MS analysis method 2 & 3 as shown in Table S9.

In order to determine [$^{19}$F]FDT metabolism in vivo, we assessed concentration vs time in two mammalian species: common marmosets and C57BL/6 mice. For marmosets weighing between

290.3–340.3 g, 20 mg of sterile-filtered FDT in PBS was injected IV ($n = 3$) and ~200 μL blood samples were taken from the femoral vein at 0 min, 5 min, 30 min, 60 min, 90 min and 180 min post injection. Samples were spun (4000 rcf for 10 mins at room temp) to separate red cell mass from plasma. Plasma portion was separated and stored at −80 °C for further analysis. Similarly, mice weighing from 18.7–20.5 g were injected ($n = 5$) with 5 mg of filter-sterilized FDT solution dissolved in PBS solution, pH 7.4. Mandibular blood samples were taken at 2 min, 15 min, 30 min and 45 min post injection. For the last sample in each mouse, the animal was bled out to get maximum sample for NMR analysis. Final urine samples were collected in 3 out of 5 mice through free catch. Blood samples were centrifuged (6500 rcf for 4 mins at room temp) to separate red cell mass from plasma, and plasma was separated and stored at −80 °C prior to analyzes.

Plasma samples from −80 °C were thawed on ice and mixed with 80% acetonitrile to precipitate plasma proteins and other biomolecules. Samples were centrifuged at 10,000 g, 4 °C for 10 min. Supernatant was removed and analyzed by LC-MS in a SIM mode using AdvancedBio glycan column 2.7 μm, 4.6 × 250 mm, flow rate 0.5 mL/min and gradient elution of acetonitrile (solvent A) and water with 50 mM ammonium formate, pH 4.5 (solvent B); gradient composition: 0 min – 10% B, 15 min – 38% B, 25 min – 38% B, 35 min – 38%B, 36 min – 10% B and 46 min – 10% B. Similar to samples, FDT calibration curve was generated and amount of FDT present in the samples was quantified using the calibration curve data.

### PET/CT instrumentation, probe administration, and image acquisition

Rabbits and marmosets at NIAID were anaesthetized and maintained during imaging[49,67]. Syringes of FDG or FDT were measured in a dose calibrator immediately before and after injection, targeting = injected doses of 2 mCi/kg. During uptake and distribution of the probes, a CT scan from the base of the skull spanning the lungs and the upper abdominal cavity was acquired as described for each species on a helical eight-slice Neurological Ceretom CT scanner (NeuroLogicia, Danvers, MA) operated as part of a hybrid pre-clinical PET/CT system utilizing a common bed. The animal bed was then retracted into the microPET gantry (MicroPET Focus 220, Siemens Preclinical Solutions, Knoxville, TN) and sixty minutesβ ± 5 min post FDG injection, a series of 2 or 3, 10 min emission scans with 75 mm thick windows with a 25 mm overlap were acquired caudal to cranial. The FDT scans were acquired beginning at 60, 90, and 120 min ± 5 min post injection with the same duration, window and overlap. For the blocking studies, injections of 150 μg of [19F]FDT synthesized as described above were administered 60 min and 5 min prior to the [18F]-FDT tracer and the scans were acquired as before. Two animals were used in the blocking studies with one animal receiving the blocking agent while the other was administered saline only prior to the FDT tracer. Two days later, the administrations were reversed so that the second animal received the blocking agent. The emission data for all scans were processed and corrected[67]. Cynomolgus macaques ($n = 3$) at the University of Pittsburgh were imaged using a Siemens microPET Focus 220 PET scanner and a Neurological Ceretom CT scanner[54]. Paired FDG and FDT scans were collected on only 2 macaques pre-necropsy. Scans were viewed using OsiriX (Pixmeo, Geneva, Switzerland) to identify and analyze individual granulomas as described previously[68]. Uptake of FDG in the myocardium was variable among individuals and scan dates as has been noted previously[69].

### Uptake/time course analysis PET/CT data analysis

For FDG and FDT activity measurements, PET/CT images were loaded into MIM fusion software (MIM Software Inc, Cleveland, OH USA) to create lung contours using the CT 3D region growing application with upper and lower voxel threshold settings of 2 and −1024 HU respectively with hole filling and smoothing applied. Dense lesion centers were subsequently identified for inclusion in the lung region manually and the program calculated the FDG signal parameters. In addition, each lesion within the lung was marked with a 3-D region of interest (ROI). In uninfected marmosets administered FDT, ROIs with a similar-volume (dia. 5 to 7 mm) were distributed in the lung as mock lesions according to the method of Ordonez et al.[70] and the SUV statistics for the ROIs were captured into excel sheets for analysis[67].

### Biodistribution of [18F]FDT in Rhesus and Marmosets

An adult male rhesus macaque (12.4 kg) was administered 151 MBq of [18F]FDT via saphenous vein catheter after being anesthetized with ketamine and then maintained with 2–4% isoflurane/97% oxygen. Dynamic whole-body PET images were obtained on a Siemens mCT PET/CT scanner. The images were reconstructed using an iterative time-of-flight algorithm with a reconstructed transaxial resolution of about 4.5 mm. The PET acquisition was divided into 22 time frames of increasing durations, 2 × 15 sec, 4 × 30 sec, 8 × 60 sec, and 8 × 120 sec. Each frame in the sequence was gathered from 4 bed positions to obtain whole body dynamic data over the scan duration (see Fig. 7A). Volumes of interest (VOIs) were drawn over major organs, non-decay corrected time-activity curves were generated and integrated, and then used to generate organ residence times that were scaled to human body and organ weights. Organ radiation absorbed doses and the effective dose were then calculated using OLINDA 2.1 (Vanderbilt University, Nashville TN)[71,72].

Two marmosets were administered 2 mCi/kg FDT (in ~300 μL PBS) via tail vein catheter while anesthetized with 3% isoflurane/97% oxygen and PET/CT scans were collected as described above and a final 200 μL whole blood sample was drawn via femoral vein puncture and placed immediately into a 1.5 mL tube that was coated with potassium ethylenediamine tetraacetic acid. After euthanasia, the liver, right kidney, skeletal muscle (right quadriceps), brain, and bone (femur) and other organ samples were harvested and weighed. Each of the tissue samples were transferred to gamma counter tubes and counted on a Wallac Wizard 1480 Gamma Counter (Perkin Elmer, Waltham, MA). Count data were converted to percent injected dose per gram using a dose calibration curve.

### Metabolism analysis [18F]FDT or 18F-labeled analogs

Blood (100 μL), urine (100–300 μL) and lesion samples were collected from animals and diluted 2:1 v/v with acetonitrile to sterilize them. These samples were then vortexed, centrifuged and the supernatant analyzed by either HPLC or TLC. The TLC silica gel plates used to assay these metabolism samples and lesion extracts were 150 Å Silica Gel HL 250 μm 10 ×20 cm channeled plates with a pre-adsorbent zone purchased from Analtech/iChromatography (Newark, DE). TLC plates were visualized via exposure to a phosphor screen for 16 h prior to imaging using a phosphor screen and Typhoon FLA7000 (GE Healthcare, Pittsburgh PA) or using a BioScan AR-2000 reader.

### 14C-labeled-[19F]FDT incorporation into Mtb lipids

For the generation of [U-14C,19F]FDT, in a 15 mL falcon tube, ATP (Sigma Aldrich) 29 mg (52 μmol) was added and dissolved in 400 μL of HEPES buffer solution (HEPES 100 mM, pH 7.4, MgCl₂ 10 mM and NaCl 200 mM). The pH of the solution was also adjusted to 7.4. 5 mg hexokinase from yeast (Sigma) were added to the solution, to which 150 μL of [U-14C]2-FDG (American Radiolabelled Chemical, Inc., 50 μmol, 15 μCi, 300 μCi/mmol) was then added and incubated without agitation for 1 h at 37 ˚C, 100 rpm. After 1 h, UDP-Glc (34 mg, 60 μmol), OtsA enzyme (2 mg) and OtsB enzyme (1 mg) were added and reaction volume adjusted to 1 mL using HEPES buffer solution and reaction mixture incubated at 37 ˚C, 100 rpm for 15 h. The mixture was then

heated at 95 °C for 10 min, centrifuged (3000 rpm, 10 min) and the supernatant transferred into a fresh 15 mL falcon tube. Samples were analyzed by TLC for the formation of [U-$^{14}$C,$^{19}$F]FDT using TLC silica gel plates as a stationary phase (EtOAc:iPrOH:H$_2$O, 4:4:1 as mobile phase). Plates were visualized via phosphor imaging as above.

For analysis of [U-$^{14}$C,$^{19}$F]FDT incorporation into *Mtb* lipid, 200 mL Mtb culture was grown in 7H9/AGDN/Tween 80 media at 37 °C until OD$_{600}$ ~ 0.5–0.6. Cells were harvested by centrifugation (3500 rpm, 10 min). Supernatant was discarded and the pellet was re-dissolved in 4 mL of same media. [U-$^{14}$C,$^{19}$F]FDT (750 μL, 15 μCi) was added into the culture and incubated for further 24 h. Cells were then harvested by centrifugation (3500 rpm, 10 min), washed with PBS buffer (100 mM PBS, pH 7.4). Lipids were extracted using 5 mL of 2:1 chloroform:methanol by vigorously shaking overnight. After filtering, the filtrate was evaporated using a stream of argon gas. The resulting residue was re-dissolved in 80:20 chloroform:methanol and 30 μL of sample was analyzed by TLC as above using TLC silica gel plates as a stationary phase (chloroform:methanol:water, 90:9:1 and chloroform:methanol:water 75:25:4 as mobile phase, Fig. S3F).

### Human Microsomal Stability Assay

Microsomal stability assay was carried out as described previously[73]. In brief, 10 x reduced nicotinamide adenine dinucleotide phosphate (NADPH)-regenerating system was freshly prepared by dissolving 37.2 mg glucose-6-phosphate (Sigma) in 1 mL of 100 mM potassium phosphate buffer, 39.8 mg nicotinamide adenine dinucleotide phosphate (NADP, Sigma) in 1 mL of potassium phosphate buffer, 11.5 U glucose-6-phosphate dehydrogenase (Thermo) in 1 mL of potassium phosphate buffer and 26.8 mg MgCl$_2$.6H$_2$O (Sigma) in 1 mL of distilled water. All these solutions were combined and kept on ice. For verapamil analysis, 263 μL phosphate buffer (100 mM, pH 7.4), 16 μL freshly thawed pooled human microsomes (Sigma) and 3 μL of verapamil (1 μM final concentration) were incubated at 37 °C for 5 min. After 5 min, 18 μL of NADPH-regenerating system was added to initiate reaction. Immediately after adding the NADPH regenerating system, 50 μL, 0 min time point sample was taken and quenched with equal volume of ice-cold methanol. Similarly further samples were taken at 10 min, 30 min, 60 min, 90 min and 120 min. Samples were centrifuged at 10,000 g at 4 °C for 10 min. The supernatant was analyzed by LC/MS under single ionization mode (SIM) mode using poroshell EC-C18 column (2.7 μm, 2.1 × 50 mm), flow rate 0.3 mL/min (Agilent) and gradient elution of water with 0.1% formic acid (solvent A) and Acetonitrile with 0.1% formic acid (solvent B) over 21 min, where gradient elution composition was 0 min – 5% B, 1 min – 5% B, 11 min – 95% B, 15 min – 95% B, 17 min – 5% B and 21 min – 5% B. For [$^{19}$F]FDT, 529 μL potassium phosphate buffer, 32 μL human, pooled liver microsomes (stock 10 mg/mL) and 3 μL of FDT (final concentration 3 μM) were used. After incubating the reaction mix at 37 °C for 5 min, 36 μL of NADPH regenerating system was added to initiate the reaction. 50 μL samples were taken at 0 min, 2.5 min, 5 min, 10 min, 20 min, 40 min and 80 min and reactions were quenched by adding equal amount of ice-cold acetonitrile and kept on ice. Samples were then centrifuged at 10, 000 g at 4 °C for 10 min and supernatant was analyzed by LC-MS under SIM mode using AdvancedBio glycan column (2.7 μm, 4.6 ×250 mm), flow rate 0.5 mL/min and gradient elution of acetonitrile (solvent A) and water with 50 mM ammonium formate, pH 4.5 (solvent B); gradient composition: 0.01 min – 32% B, 15 min – 38% B, 20 min – 32% B and 25 min – 32% B.

### Toxicity studies

For toxicity studies, 2 g of [$^{19}$F]-FDT was synthesized and tested for quality control as described above and in Figs.39 and shipped to a contract research organization. Toxicity studies were conducted by SRI International, Biosciences Division, Menlo Park, CA, USA in rats and

dogs. The experimental design is described in Tables S10, S11, respectively.

For studies in rats, male and female Sprague Dawley rats (15/sex/group) were given a single iv administration of [$^{19}$F]-FDT at 1.32 mg/kg (100 times of human dose; Group 3) or 13.2 mg/kg (1000 times of human dose; Group 4) on day 7 or 1.32 mg/kg/day for 7 consecutive days (total dose 9.24 mg/kg; Group 2. A control group (15/sex; Group 1) was given 7 days of vehicle (50 mM HEPES buffer, pH 7.5 with 7.5 mM MgCl$_2$) at an equivalent dose volume of 1 mL/kg on days 1–7. Animals were euthanised on day 9 (main groups) or day 21 (recovery groups).

To assess the toxicity associated with [$^{19}$F]-FDT in dogs, male and female Beagle dogs (5/sex) were given a daily iv injection of [$^{19}$F]-FDT at 0.4 mg/kg/day (100 times of human dose; Group 2) for 7 consecutive days, or a single iv administration of [$^{19}$F]-FDT at 0.4 or 4 mg/kg (100 and 1000 times of human dose in Groups 3 and 4, respectively) on Day 7. A control group (5/sex; Group 1), was given 7 days of vehicle (HEPES buffer, pH 7.5, 50 mM with MgCl$_2$ 7.5 mM), at an equivalent dose volume of 0.25 ml/kg on Days 1–7. Animals were sacrificed on Day 9 (3/sex/ group; main groups) or Day 21 (2/sex/group; recovery groups). The single dose administration for Groups 3–4 were initiated on Day 7 so that clinical pathology and necropsy occurred on the same calendar day for all Groups (1–4), thus allowing for sharing of vehicle control data for analysis of clinical pathology and necropsy results between the two dose regimens.

### Reporting summary

Further information on research design is available in the Nature Portfolio Reporting Summary linked to this article.

## Data availability

Raw NMR spectra, small molecules and protein LC-MS raw data, sequence files and radio-HPLC raw data has been deposited in the Zenodo repository (https://doi.org/10.5281/zenodo.10624807). PET-CT data related to infected and naïve marmosets has been deposited into Accessclinical data@NIAID with the link: https://accessclinicaldata.niaid.nih.gov/study-viewer/clinical_trials/MARM-TB-FDT. Source data are provided with this paper.

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

## Acknowledgements

We thank Becky Sloan for technical assistance with animal procedures and the Comparative Medicine Branch (NIAID) for animal handling. We thank the NIH PET Department for providing excellent technical support. We thank James Collier from the Nasmyth group (Biochemistry, University of Oxford). and Shubhashish Mukhopadhyay from the Burgess-Brown group (NDM, University of Oxford). for their assistance in larger-scale enzyme expressions in insect cells. We also thank Oxford Chemistry Department NMR and Mass Spectrometry teams for their continued support in experimentation and data analysis. Supplementary Figs. 11 (panels G,H), 35 (panel A) and 36 (panel B), created with BioRender.com, released under a Creative Commons Attribution-NonCommercial-NoDerivs 4.0 International license. This research was supported by the Intramural Research Programs of the NIAID, NHLBI, CC and NIBIB, the CAPS program of NIH (1ZIAAI001164 to CEB), the EPSRC University of Oxford Impact Acceleration Account (EP/R511742/1 to BGD) and the Bill and Melinda Gates Foundation (OPP1034408 to JLF, CEB, BGD). The content of this publication does not necessarily reflect the views or policies of the Department of Health and Human Services, nor does the mention of trade names, commercial products, or organizations imply endorsement by the U.S. Government. The Chemistry theme at the Rosalind Franklin Institute is supported by the EPSRC (V011359/1(P) to BGD).

## Author contributions

D.M.W., M.L.S., D.M.S., E.D., P.H., L.E.V., J.L.F., C.A.S., J.A.T., L.J.F., and C.E.B. designed and implemented the various animal studies; R.M.N.K., Y.M.A., G.A.M., F.B., R.S., F.D'H., D.O.K., D.A., B.G.D., and N.S.M. designed and performed the radiochemistry; R.M.N.K., F.D'H., S.S.L., D.A., and B.G.D designed and implemented the various cold chemistry experiments; R.M.N.K. and N.Y. established the insect cell expression system via virus bacmid DNA construction, virus amplification, and initial expression and purification; R.M.N.K., S.S.L., Y.G., and R.R. expressed enzymes and performed kinetic analyzes; K.M.B. performed initial syntheses and proof-of-principle experiments in prototype probes; W.D., M.K.P., F.G., and A.G.W. analyzed PET/CT scans; R.M.N.K., L.E.V., B.G.D., and C.E.B. drafted the manuscript; all authors approved the final version of the manuscript.

## Competing interests

The authors declare no competing interests.

## Additional information

R.M. Naseer Khan [1,13,15], Yong-Mo Ahn[2,15], Gwendolyn A. Marriner[2,15], Laura E. Via [2,3,4,15], Francois D'Hooge[1], Seung Seo Lee [1,5], Nan Yang[1,6], Falguni Basuli[7], Alexander G. White[8], Jaime A. Tomko[8], L. James Frye[8], Charles A. Scanga [8], Danielle M. Weiner[2], Michelle L. Sutphen[2], Daniel M. Schimel[2], Emmanuel Dayao[2], Michaela K. Piazza[3], Felipe Gomez[3], William Dieckmann[9], Peter Herscovitch[9], N. Scott Mason[10], Rolf Swenson[7], Dale O. Kiesewetter[11], Keriann M. Backus[1,14], Yiqun Geng [1], Ritu Raj[1], Daniel C. Anthony [12], JoAnne L. Flynn [8], Clifton E. Barry III [2,4] ✉ & Benjamin G. Davis [1,6,12] ✉

[1]Department of Chemistry, University of Oxford, Chemistry Research Laboratory, Oxford, UK. [2]Tuberculosis Research Section, Laboratory of Clinical Immunology and Microbiology, Division of Intramural Research (DIR), National Institute of Allergy and Infectious Disease (NIAID), National Institutes of Health (NIH), Bethesda, MD, USA. [3]Tuberculosis Imaging Program, DIR, NIAID, NIH, Bethesda, MD, USA. [4]Institute of Infectious Disease and Molecular Medicine, University of Cape Town, Cape Town, South Africa. [5]School of Chemistry, University of Southampton, Southampton, UK. [6]The Rosalind Franklin Institute, Oxfordshire, UK. [7]Chemistry and Synthesis Center, NHLBI, NIH, Bethesda, MD, USA. [8]Department of Microbiology and Molecular Genetics, University of Pittsburgh, Pittsburgh, USA. [9]Positron Emission Tomography Department, Clinical Center, NIH, Bethesda, MD, USA. [10]Department of Radiology, University of Pittsburgh, Pittsburgh, USA. [11]Molecular Tracer and Imaging Core Facility, National Institute of Biomedical Imaging and Bioengineering, NIH, Bethesda, MD, USA. [12]Department of Pharmacology, University of Oxford, Oxford, UK. [13]Present address: Clinical Pharmacology Lab, Clinical Center, NIHBC, NIH, Bethesda, MD, USA. [14]Present address: Biological Chemistry Department, David Geffen School of Medicine, UCLA, Los Angeles, CA, USA. [15]These authors contributed equally: R. M. Naseer Khan, Yong-Mo Ahn, Gwendolyn A. Marriner, Laura E. Via. ✉e-mail: CBARRY@niaid.nih.gov; Ben.Davis@rfi.ac.uk

