## [Peer Review File · Nature Communications]

Distributable, Metabolic PET Reporting of TuberculosisReviewer #1 (Remarks to the Author):

This manuscript utilises chemoenzymatic methods to synthesise 2-[¹⁸F]fluoro-2-deoxytrehalose ([¹⁸F]-FDT) using 2-[¹⁸F]fluoro-2-deoxy-D-glucose ([¹⁸F]FDG) as the substrate. They identify the best radio-analogue as well as the synthetic route to perform this practically. The authors also provide extensive data from animal studies on the application of this new imaging method, which if successful would be of major medical importance.

1. The authors state that their yield "was essentially dependent on [¹⁸F]FDG source". Since many commercially available [¹⁸F]-FDG sources have poor specific activity, can you authors provide details on their studies and also state the minimum specific activity [¹⁸F]FDG source tested, that yielded adequate [¹⁸F]-FDT activity. Similarly, can the authors provide clarity on whether activities of >20 mCi [¹⁸F]-FDT (needed per patient for a clinical study) are being routinely produced with their method.

2. [¹⁸F]-FDT specific activity of 69 ± 26 mCi/mg (23.6 ± 8.8 Ci/mmol) is on the lower side and as it stands, a standard dose of 10-20 mCi per patient will exceed the upper limit of the microdosing range (<100 µg) for PET studies per the EU guidelines (and likely other countries). This may complicate the regulatory approvals for clinical translation.

3. Trehalose uptake may not specific to Mycobacterium tuberculosis. In fact, trehalose metabolism occurs in a wide range of microorganisms, including bacteria, fungi, etc. For example, Corynebacterium, a common upper respiratory commensal as well as other commensal mycobacteria, and fungi may take up trehalose. Therefore, it would be important to: a) test the in vitro ¹⁸F-FDT uptake by common pathogenic and commensal bacteria; b) remove / clarify the claim in the manuscript (abstract and elsewhere) that ¹⁸F-FDT has Mtb specific processing.

4. Data on metabolism of [¹⁸F]-FDT in rabbit and NHPs is provided, but metabolism can be species specific and so data from in vitro human liver microsomal assays would be important.

5. Only cross-sectional images are shown for PET/CT images in Figures 3-6. However, whole body views including the heart, liver and kidneys are needed for these figures. Additionally, CT, PET and fused PET/CT images should be shown separately for the areas of interest.

6. The control for Figure 3 (panels A and B) are "naïve animals". However, sterile inflammatory controls, which are critical (as the patient would be presenting with some illness that needs to be distinguished from TB) are not shown anywhere in the manuscript. Additionally, infections with other pathogens (e.g. bacterial pneumonias) are not shown as a control. These controls (sterile inflammation and other pathogens causing pneumonias) are needed to assess the specificity of the imaging approach.

7. Blocking studies: The decrease in the PET signal in the blocked animals is not substantially lower versus unblocked animals – can the authors explain this? Additionally, blocking seems to "block" a likely non-specific uptake (gall bladder - bright lesion in the liver, Figure 3E). If this is accurate, can the authors explain this? Whole body images (including other cuts) would be helpful to understand the anatomic location of this signal. Finally, it seems from the methods that only two animals with cross-over design (although the figure 3 legend states that 22 animals were used, which seems like a typographical error as it seems that there are less than 22 points on the graph), or a total of two blocked and two unblocked animals, were used for these studies. The authors need to clarify the number of animals used and if it is only 2, that is a small sample size and additional studies should be performed.

8. Figure 4 is difficult to interpret. Can the authors provide whole body images / 3D images and objective measures from relevant regions of interest?

9. The differences in the [¹⁸F]-FDT and [¹⁸F]-FDG PET images are difficult to interpret visually in Figure 5 (panels A-B, E-F). Also, the final bacterial burden is provided for the lesions, but what is the expected bacterial burden at the start of treatments? Given that individual lesions within the same animal can have highly variable PET activity as well as bacterial burden (some can be

sterile), it would be important to know the number of animals imaged and a large sample size is needed to demonstrate the effect of treatment due to this variability.

10. In figure 6, [18F]-FDT PET (panel A) shows high uptake in heart and the liver. Why is that? Conversely, [18F]-FDG PET shows much lower activity in the heart and liver, which is unusual as the myocardium takes up [18F]-FDG. Can the authors explain these data?

11. Can the authors specify the number of animals used for the studies shown in Figure 7. In fact, all figure legends should clarify the number of animals used for each study, which is currently unclear. For some studies the sample size is small (e.g. $n = 2$) and therefore it is difficult to draw definitive conclusions.

Other comments:

12. Reference 1 can be updated to the 2022 report.

13. Page 18, second para: "pre-blocking' experiments were conducted with both trehalose and 'cold' [19F]FDT (administered 1 h and 5 minutes)". Based on the figure 3 legend it seems that the 'cold' [19F]FDT was administered at 1 h as well as 5 minutes prior to the PET studies, but this is unclear in the text. Please clarify why blocking agent was administered twice, which should also be clarified in the results section.

14. What is the expected limit of detection of this PET imaging approach?

15. Could the authors provide a summary of the primary data (clinical pathology / lab data, etc.) for the toxicology studies?

Reviewer #2 (Remarks to the Author):

Khan et al report the synthesis and characterization of 18F-labeled trehalose (18F-FDT) as a candidate for PET reporting of Mycobacterium tuberculosis (Mtb) infection in vivo. The importance of such a technology, should it be clinically successful, is clear: tuberculosis is a disease that afflicts 10 million people and kills 1.5 million people on an annual basis, and there is not currently a way to accurately and non-invasively image active disease progression and response to therapy in infected individuals. Although the standard PET tracer 18F-FDG can be used for TB imaging, it reports on host inflammation rather than metabolically active Mtb, and this approach can be confounded by non-TB lung pathologies. The idea here is to develop a TB-selective tracer based on 18F-labeled trehalose. Trehalose is a disaccharide that Mtb enzymatically incorporates into surface lipids. The idea is founded on a strong body of literature demonstrating that various trehalose derivatives can enzymatically incorporate into Mtb.

This is a sweeping, tour de force study inclusive of synthesis, enzymology, animal models, disease imaging, and tracer production/safety. The scope of the study is remarkable and its potential impact is exceptional, as noted above. Most of the key assertions about this technology are supported by the data presented. There are some issues described below that need to be addressed. Comments are given below in categories of tracer synthesis, labeling characterization, and imaging/safety studies, and additional minor comments are given at the end.

Synthesis

Chemical/chemoenzymatic synthesis routes to four different FDT isomers with fluorine at different positions were developed to enable a comprehensive analysis and selection of the most promising PET tracer candidate. This approach proved useful, as careful analyses revealed that one of the isomers could not be efficiently accessed chemoenzymatically by the reported method (3-FDT) and two were prone to degradation in serum or in vivo (6-FDT and epi-4-FDT). 2-FDT was identified as

the most promising isomer. A major strength of the work is that 2-FDT can be accessed directly from the standard PET tracer FDG. This was done using a robust one-pot enzymatic synthesis process employing hexokinase, OtsA, and OtsB enzymes that were produced in high yield without endotoxin present. It was clearly demonstrated that the carefully optimized method could generate very large amounts of the cold analog 19F-FDT and reliably produce the hot analog 18F-FDT from FDG in good radiochemical yield (~40%) and high purity. The characterization of FDT structure and purity was impeccably done.

There are several issues with the synthetic work that need to be addressed:

- Four different enzymatic systems were tested for synthesis of FDT starting from FDG, including the three-enzyme system noted, a two-enzyme system using an OtsAB fusion protein, and two different TreT enzymes which can convert FDG to FDT in a single step. Various parameters were assessed to decide which system to proceed with. One issue was that the enzyme kinetics experiments (Table 1) measured only native glucose/G6P substrates as the acceptors, rather than FDG/FDG6P that would be used in FDT synthesis. Thus, the data do not establish which of these systems has the best activity for FDT synthesis. Kinetic analysis of FDG (most importantly 2-FDG) would need to be done in order to provide the most relevant comparison, otherwise the limitation of the experiments done should be noted.
- Related to the above, there are literature reports that used TreT enzymes from *T. tenax* and *T. uzoniensis* to synthesize cold 19F and hot 18F-FDT from FDG in one step, which should be discussed (PMIDs: 27560008, 30428395, 32953231). This includes a one-step one-enzyme radiosynthesis of FDT from FDG done by TreT from *T. tenax* in 30 min and 70% radiochemical yield (PMID 30428395). In addition, the TreT enzymes can efficiently make 3-position and 6-position FDT and the *T. uzoniensis* enzyme can make the 4-position FDT (PMID: 27560008 and 32953231), so the isomer versatility is good. Therefore, the three-enzyme system using OtsAB benefits from excellent pyrogen-free enzyme production yields and the ability to scale up FDT synthesis. On the other hand, the one-enzyme TreT systems benefit from one-step synthesis, isomer flexibility, and efficient FDT syntheses, but the enzyme production yields are lower. A more thorough and balanced consideration of methods and their advantages and disadvantages is warranted.
- Figure S9 shows that the TreT enzymes had modest conversions of FDG to FDT (10-40%). However, these results are inconsistent with the literature reports from independent groups, in which quantitative conversions were obtained in ≤1 h (papers noted above). Compared to the literature reports, in this study ~1/10th the TreT enzyme was used (Table S6) and the reactions were run at sub-optimal temperatures (37 C) (Page 18 of SI). These enzymes are thermostable and reported reaction temperatures are rather run at 60-70 C, at which enzyme activities are much higher (PMID: 27560008). This is likely why the results were inconsistent.
- Given the above points, the conclusions about the synthesis ("higher biocatalyst expression yields, flexibility of enzyme usage (differential loadings), stability and superior kinetic parameters, which, together, gave rise to more efficient synthesis in our hands") should be revisited.

Characterization of labeling

There were experiments done to characterize metabolic stability of the various FDT isomers. It is clear that 6-FDT and (to a lesser extent) epi-4-FDT have some amount of breakdown (Figure S3, note that captions for panels C and D are opposite figures, please fix). This in part led to judicious focusing on 2-FDT, which Figure S3G shows had higher stability. However, there were some issues with the other data presented on this topic.

- Figures S35 and S36 also aim to show metabolic stability of FDT. In S35, it is not clear how long and at what temperature the samples were incubated in plasma. Also, were multiple time points assessed? (note: there are no letters on the figure corresponding to the caption).
- In S36, the sample was only taken at 5 min and the measurement appears to be a qualitative assessment of whether FDT is present. It would seem difficult to conclude that there is no degradation of the tracer from this experiment. Is there a plot for 2-FDT similar to those for the 6-

and epi-4 compounds in Fig S3C and D that could be shown?

The experiments to characterize Mtb labeling by FDT had some issues that need to be addressed. Most significantly, there were no in vitro uptake studies done in cultured Mtb cells. This is a relatively simple and inexpensive experiment that should be a precursor to in vivo imaging, and it provides an opportunity to investigate uptake mechanism in a more straightforward manner. For example, in the cited study on 18F-sorbitol, in vitro uptake in a panel of different bacteria and mammalian cell lines was done prior to in vivo imaging of infection (PMID: 25338757). In the above-noted study on 18F-FDT, in vitro uptake in *M. smegmatis* and mammalian cell lines was done (PMID: 30428395). However, in vitro uptake of FDT in Mtb has not been previously demonstrated. Ideally, this experiment would be conducted and data shown, preferably also comparing uptake in other relevant bacterial species and potentially in mammalian cell lines. FDT is described in the paper as a TB-selective tracer, but selectivity against other types of bacteria is not demonstrated. This is of high interest in establishing the utility of FDT for TB imaging, as many types of bacteria have trehalose transporters and trehalose-metabolizing enzymes.

The possible mechanism of FDT incorporation was investigated through TLC analysis of putatively labeled lipids from ex vivo or in vivo FDT treatment experiments. The identification of 18F-labeled trehalose mycolates would support a mechanism in which FDT is lipidated by antigen 85 as shown in Figure 1. Based on the TLC analysis shown (Figure S3E and F), it is claimed that radiolabeled TMM and TDM are observed, but the data as presented do not clearly support the conclusion:

- The TLC lanes are not labeled so it is unclear what standards/conditions are being presented.
- The annotations on the TLC plates are confusing. In some cases, the arrows do not clearly point to a spot, and for the epi-4-FTre label, is this referring to the unmodified tracer?
- Why was epi-4-FDT used for these experiments instead of the 2-FDT tracer that is the main focus of the manuscript?
- The main text says authentic standards were compared to assist in identifying spots of interest, but these data/comparisons are missing. It is difficult to judge whether the spots shown are likely to represent labeled lipids without standards shown.
- The extraction procedure and TLC conditions were not described in detail. In the main text methods, there is a section on "detection of labeled mono and dimycolates" but it only describes analysis of blood and urine samples and does not seem relevant, whereas the Figure S3E/F caption refers to lung samples and gives very little detail. The SI methods did not appear to cover these experiments. It is very challenging to interpret the data shown without more experimental detail.

In addition, in vitro experiments to investigate mechanism of uptake would be valuable. Presumably, extraction and analysis of labeled lipids from Mtb grown in vitro would be a good first step, prior to going into infected animals which significantly complicates the analysis. Given that FDT incorporation into mycolates has never previously been demonstrated, this would be a good experiment, particularly in light of the at-present questionable ex vivo/in vivo lipid labeling experiments.

Another question related to characterization of mechanism is if the authors considered whether the Mtb trehalose transporter could be involved in uptake of the tracer, as previously demonstrated in *M. smegmatis* (PMID: 27560008). This is of interest because if the tracer is solely processed through antigen 85 that would lend toward mycobacteria-selective uptake and a TB-selective tracer. On the other hand, if uptake is driven by trehalose transport, there could be selectivity issues since as noted above this is a common feature of many bacteria (albeit there are different transporter types with likely different specificities for trehalose derivatives). Discussion of the potential involvement of trehalose transport systems in FDT uptake would be valuable.

Imaging and safety

The PET imaging experiments, conducted in multiple animal infection models, strongly support FDT

as a tracer for monitoring Mtb burden and response to antibiotic treatment in vivo, which as noted above could be a powerful preclinical and potentially clinical capability. Some questions on the imaging experiments are given below:

- The comparison to TB imaging using FDG demonstrated differential labeling patterns in the replicate shown. The n=3 for this experiment, can images from additional replicates be added to the SI? Also, if there were uninfected control animals for this experiment, those would be useful to show in Figure 4 and/or the SI.
- It was observed that uptake of FDT reached maximum signal to noise at 90 min, how does this compare to FDG and, if different, please comment? Was FDT/FDG uptake monitored at earlier time points than 60 min, and if so, how did overall signal and distribution compare?
- Figure 7B/C shows relatively little FDT in lungs, can the authors comment on the consequences of this for imaging pulmonary TB? It would also be interesting for the authors to comment on how the other FDT biodistribution data may impact potential imaging of extrapulmonary TB.

The safety studies, conducted in multiple animal types and enabled by the excellent scaled up cold FDT synthesis, demonstrated no adverse effects, setting the stage for subsequent clinical testing in humans.

Additional minor comments and edits

- Title: The meaning of "distributable" may be somewhat ambiguous to some readers as to whether this means the technology is distributable to (pre)clinical facilities or the tracer is distributable to different tissues in vivo.
- Abstract: "can act as a mechanism-based enzyme reporter in vivo" is a bit confusing, perhaps "mechanism-based reporter of Mtb-specific enzyme activity in vivo" or something to this effect?
- Abstract: suggest "custom-made" instead of "bespoke"
- Page 3: Cite WHO 2022 report/data.
- Page 3: The sentence beginning "The analysis⁴ and internationally-agreed, comprehensive monitoring of access to PET-CT" is difficult to follow, please rephrase.
- Page 4 typo: "...and so only can only help..."
- Page 4: Cord factor refers to TDM, not TMM (sentence is ambiguous).
- Page 4: Ag85 is generally considered to be cell envelope-associated or secreted, but to this reviewer's knowledge is not commonly thought to, or depicted to, significantly associate with the plasma membrane as shown in Figure 1a.
- Page 5: "could function as both highly specific and sensitive reporters" (remove "a")
- The authors should comment on why epi-4-FDT rather than 4-FDT was targeted, e.g. synthetic accessibility or other reason.
- Can authors comment on the low yield (15%) of fluoride substitution en route to 4-epi-FDT?
- Page 9 define RCY (this is given later); no need to capitalize "one pot"
- SI P56, Figure S9 caption, give FDG substrate amounts as concentrations rather than ug.
- Page 16 Figure 2B, give FDG substrate in concentration
- Comment on why such a large excess of UDP-Glc was required for optimal conversion (130-260 mM), and whether that led to any purification issues?
- Figure 2C, is this monitoring over time? If so, show time durations on figure.
- Page 14, "consistently and repeatedly."
- Page 15, define GMP on first use.
- Page 16 Figure 2D and associated text, can the authors clarify exactly what is meant by batch? Are these just repeats of the reaction with fresh enzyme/reactants, are enzymes being re-used, are they from different protein prep batches?
- Figure S33, caption says (for a-c) and for d, but it is not clear what these letters refer to.
- Page 30 "scales of up to grams" statement clearly refers to 19F-FDT but occurs immediately after referring to 18F-FDT.
- Page 33, should be Michaelis-Menten
- Since a significant number of small errors and typos were noted, the authors are strongly encouraged to carefully check and edit the main text and SI thoroughly.

Reviewer #3 (Remarks to the Author):

The authors of this interesting work investigated pathogen-specific imaging of tuberculosis with radio-fluorinated trehalose. The authors leveraged on the specificity of trehalose utilization by Mtb for specific pathogen targeting. Different fluorinated trehalose molecules were tested for their efficiency of radiosynthesis, length of reaction time, and in vivo stability. [18F]FDT demonstrated superior performance and was investigated further. A scalable technique for synthesizing [18F]FDT was subsequently described. [18F]FDT showed rapid and specific incorporation into mycobacterial mycolic acid. In vivo biodistribution in marmoset shows no significant [18F]FDT uptake in normal lung tissues, radiotracer uptake in tuberculous lung lesions with signal reduction in blocking experiment confirming the specificity of binding, minimal increase in the intensity of radiotracer uptake in lesions at 90 minutes compared to 60 minutes post tracer injection with no further improvement in avidity beyond 90 minutes, and a direct relationship between the radiotracer avidity of the lesions and the number of culturable Mtb within the lesions. The spatial distribution of FDT uptake differs from that of FDG, reflecting the differences in the target engagement between the two tracers. Interestingly, there was a significant reduction in FDT but not FDG signal corresponding to a reduction in Mtb burden in treated marmosets, suggesting that FDT may be a useful biomarker of Tb response assessment. The safety of [18F]FDT in the experimental animals was demonstrated. In dosimetry study, low FDT uptake in normal organs was reported with the target organ being the urinary bladder wall.

FDG is the most commonly used radiotracer for infection imaging, including Tb imaging. Its lack of specificity for infection limits its clinical application. A need, therefore, exists for novel radiotracers that are specific for pathogens. The report in this manuscript of 18F-FDT is a welcome development towards achieving this goal.

Specific comments:

1. Results - [18F]FDT is also an Effective Mtb-Radiotracer in an 'Old World' Non-Human Primate: As shown in figure 6, [18F]FDT PET signal at 60 and 120 minutes were compared between lesions with culturable Mtb bacilli and those without. Why was 90 minutes, the optimum imaging time point left out here? Is there same data for FDG as well? It will be great to see how FDT compares with FDG in this regard.
2. An important finding from this study is the spatial incongruence in the uptake of FDT versus FDG in the same lesion. This may support differences in target engagement between the two tracers, however, a proof of this is necessary. This proof needs to show the concentration of Mtb bacilli in the region of FDT uptake and the concentration of inflammatory cells in the region of FDG uptake.
3. An important drawback of FDG for Tb response assessment is the ability of dead bacilli in treated Tb lesion to induce inflammatory changes causing FDG uptake in sterile lesions. FDT, as described here, has the potential to address this limitation. It will, therefore, be interesting if the authors could perform additional experiments to validate this.
4. Tb lesions in humans are highly diverse. The animal works presented here investigated the performance of FDT PET imaging in a limited spectrum of Tb lesions. In the comparative images of FDT and FDG PET as shown in figures 4 and 6, the signal from FDT imaging appears much less compared with FDG. This is, therefore, a concern regarding the performance of this novel tracer in detecting small lesions, especially in the context of the significant partial volume averaging that occurs with respiratory motion. Cavitary lesions are of particularly clinical interest, and the performance of FDT in this disease phenotype is unknown.
5. Discussion: Please make a brief mention of the comparison in organ dose from FDT as reported here versus FDG as reported in the literature.

Reviewer #1:

This manuscript utilises chemoenzymatic methods to synthesise 2-[18F]fluoro-2-deoxytrehalose ([18F]-FDT) using 2-[18F]fluoro-2-deoxy-D-glucose ([18F]FDG) as the substrate. They identify the best radio-analogue as well as the synthetic route to perform this practically. The authors also provide extensive data from animal studies on the application of this new imaging method, which if successful would be of major medical importance.

• Thank you

1. The authors state that their yield “was essentially dependent on [18F]FDG source”. Since many commercially available [18F]-FDG sources have poor specific activity, can you authors provide details on their studies and also state the minimum specific activity [18F]FDG source tested, that yielded adequate [18F]-FDT activity. Similarly, can the authors provide clarity on whether activities of >20 mCi [18F]-FDT (needed per patient for a clinical study) are being routinely produced with their method.

- The actual FDT specific radioactivity varied from a low of 8.3 to a high of 86.8 (Ci/mmol) across many batches.
- We have yet to produce more than 5mCi of FDT, which is more than sufficient for animal imaging, but not for humans. We have added the variability in FDT activity to the manuscript (line 328).

- In our judgment future needs can be addressed by working with specific commercial FDG vendors we can also resolve this by in-house FDG production with higher specific activities.

2. *[18F]-FDT specific activity of 69 ± 26 mCi/mg (23.6 ± 8.8 Ci/mmol) is on the lower side and as it stands, a standard dose of 10-20 mCi per patient will exceed the upper limit of the microdosing range (<100 μ g) for PET studies per the EU guidelines (and likely other countries). This may complicate the regulatory approvals for clinical translation.*

- As we note above, we believe that this is a soluble problem via two paths: automated in-house radiosynthesis to generate FDG with suitable starting specific activity and partnering with commercial suppliers in trial regions to enable such activities.
- Initial discussions with suppliers suggest that the commercial activities needed are feasible; many commercial doses are ‘end of run’. With negotiated prioritization, we believe that this is readily addressable that, with the good conversions we are seeing here, will readily allow a ‘microdose’ that below 100 μ g.

3. *Trehalose uptake may not specific to Mycobacterium tuberculosis. In fact, trehalose metabolism occurs in a wide range of microorganisms, including bacteria, fungi, etc. For example, Corynebacterium, a common upper respiratory commensal as well as other commensal mycobacteria, and fungi may take up trehalose. Therefore, it would be important to: a) test the in vitro 18F-FDT uptake by common pathogenic and commensal bacteria; b) remove / clarify the claim in the manuscript (abstract and elsewhere) that 18F-FDT has Mtb specific processing.*

- The reviewer is correct that other organisms are capable of utilizing trehalose, however no other organism other than non-tuberculous mycobacteria, is likely to concentrate ‘intact’ trehalose.
- We have previously shown that the analogue FITC-trehalose that we cite in the text is not accumulated by *Staphylococcus aureus*, *Pseudomonas aeruginosa*, or *Haemophilus influenza* (Backus et al 2011)).
- It should be noted that, here, we are not proposing that FDT would be used for the sole or even primary diagnosis of *Mtb*-mediated disease; there are indeed excellent existing ways. Instead, we are proposing that ^{18}F -FDT would be used in patients known to be infected with TB as a means of monitoring their response to therapy and perhaps in determining when sufficient treatment has been received.
- While fungal and non-tuberculous mycobacteria occasionally occur concomitant with infection with *M. tuberculosis* this is unusual and unlikely to interfere with studies designed to compare treatment regimens or establish treatment duration with an existing regimen.
- As suggested, we have altered the abstract and text accordingly.

4. *Data on metabolism of [18F]-FDT in rabbit and NHPs is provided, but metabolism can be species specific and so data from in vitro human liver microsomal assays would be important.*

- As suggested, we have now conducted *in vitro* human liver microsomal assays, using Verapamil as a comparison. These suggest that even after 80 min that $<40\%$ of FDT is metabolized.
- These experiments and data have now been added as Supplementary Figures S37G,H and to the Methods section (lines 995 onwards).

5. Only cross-sectional images are shown for PET/CT images in Figures 3-6. However, whole body views including the heart, liver and kidneys are needed for these figures. Additionally, CT, PET and fused PET/CT images should be shown separately for the areas of interest.

- As requested, 3D renderings that show the heart, liver and kidneys have been added to Figures 3 and 5 as well as now including sagittal views in addition to transverse.

6. The control for Figure 3 (panels A and B) are “naïve animals”. However, sterile inflammatory controls, which are critical (as the patient would be presenting with some illness that needs to be distinguished from TB) are not shown anywhere in the manuscript. Additionally, infections with other pathogens (e.g. bacterial pneumonias) are not shown as a control. These controls (sterile inflammation and other pathogens causing pneumonias) are needed to assess the specificity of the imaging approach.

- Again, we are not proposing using FDT as a means of diagnosing TB but instead to be used as a probe in patients already confirmed to have been infected with TB. While the suggestion by the reviewer would be the ideal comparison, it would require us to develop a new infectious model and use additional NHPs. There is, of course, a delicate balance in committing to the terminal use of NHPs *versus* what will be learned.

- In the phase I clinical trial that has been planned for this probe, we intend to recruit such individuals with other infections and/or sterile lung diseases like sarcoidosis (who are often enrolled on clinical protocols at the National Institutes of Health, Bethesda MD) to assess this specificity as suggested by the FDA and this reviewer.

- Here we compare FDT uptake in uninfected and infected animals shown in Figure 3 panels a-c, as is typical for other probes of infectious diseases (see e.g. Weinstein et al, *Sci Transl Med.* 2014, 6, 259ra146; *Antimicrob Agents Chemother.* 2012, 56, 6284 and Ordonez et al., *Antimicrob Agents Chemother.* 2015, 59, 642).

- We have added the Ordonez reference to the methods section to indicate the bases for our analyses in Figure 3 a-c (line 927).

- Specificity of labeling was of concern to us, and before committing NHPs to the project, we had previously established specificity of labeling by directly exposing three other organisms commonly found in the human lung to FITC-trehalose: *Staphylococcus aureus*, *Haemophilus influenzae* and *Pseudomonas aeruginosa*. None of these organisms were found to show appreciable labeling with FITC-trehalose in comparison to *M. tuberculosis* (Backus et al, *Nat Chem Biol*, 2011, 7, 228-35).

- We have now also noted this as a limitation of our study (lines 529-538): “The limitations of our study include the use of only small numbers of primates due to reduced number of scans that can be conducted with animals bearing untreated progressive tuberculosis. In addition, the efficacy of [¹⁸F]FDT in the presence of other possible lung diseases has yet to be assessed; we have previously determined that the analogue FITC-Tre does not label several other bacterial species that cause lung infections.(Backus et al, *Nat Chem Biol*, 2011, 7, 228-35) It should be noted that such studies will be required in Phase 1 clinical studies.”

7. Blocking studies: The decrease in the PET signal in the blocked animals is not substantially lower versus unblocked animals – can the authors explain this? Additionally, blocking seems to “block” a likely non-specific uptake (gall bladder - bright lesion in the liver, Figure 3E). It this is

accurate, can the authors explain this? Whole body images (including other cuts) would be helpful to understand the anatomic location of this signal.

- While many PET probes bind to a specific receptor and so blocking studies can demonstrate specificity, blocking is not 100%. This is dependent on the kinetics of binding k_{on} and k_{off} and hence KD of course. Representative examples include: folate receptor 60 to 80 % (Nucl Med Biol. 2012, 39, 864); *N*-methyl-D-aspartate receptor ~50% (J Nucl Med 2022, 63,1912) and integrin receptor (Theranostics 2011, 1, 403).
- Blocking of an enzyme requires additional consideration of turnover, of course. Therefore blocking by a cold substrate, as here with [¹⁹F]FDT, will see that substrate consumed and any blocking must also take into account effective competition. We achieved 40 % blocking by providing excess of [¹⁹F]FDT prior to injection of [¹⁸F]FDT tracer but this blocking is dependent on both turnover driven by Ag85 isoforms and pharmacokinetics. Our estimates of both at measured *in vivo* concentrations suggest that the blocking that we see is, in fact, consistent with near maximal. Taken together these data support specificity of action.
- The activity observed in the gall bladder suggests hepatobiliary as well as urinary clearance. It is not unusual for blocking studies to change the relative retention of the probe in organs, as again, this is dependent on relative clearance rates and their kinetic response (see, for example, Nucl Med Biol, 2012, 39, 864).
- As requested, a pair of 3D volume renderings from one of the two marmosets in the experiment has been substituted in Figure 3 for the transverse images found in the original figure. In addition, the individual contributions of each of the two marmosets in the experiment has been indicated by symbol colours (black and blue) to allow greater granularity in analysis of the data.

Finally, it seems from the methods that only two animals with cross-over design (although the figure 3 legend states that 22 animals were used, which seems like a typographical error as it seems that there are less than 22 points on the graph), or a total of two blocked and two unblocked animals, were used for these studies. The authors need to clarify the number of animals used and if it is only 2, that is a small sample size and additional studies should be performed.

- We thank the reviewer for observing the typo in Figure 3 (!); indeed, only two animals were used in the crossover design. This has been corrected (Figure 3 legend).
- While blocking studies using mice may use three or more animals, using two NHPs for such a study is not unusual given the ethical considerations (see, for example, J Nucl Med, 2022, 63, 1912; Acta Pharm Sin B, 2023, 13, 213; Pharmacol. Res., 2023, 189, 106681; Nucl Med Biol, 2007, 34, 153).
- Notably, in the example studies we cite, animals are not experiencing disease and yet specific blockers were used in only 1-2 NHPs each. *Mtb*-infected marmosets experiencing progressive TB are fragile and can become moribund quickly. In our usual model, animals are imaged after fasting once in two weeks; lesion- and disease- burden change considerably in that time frame. To conduct the blocking and reproducibility experiments (Figure 3g), it was necessary for the animals to be sedated twice in 48hrs for greater than 2hrs each day to capture the disease in the same state. Detailed consideration was given to the protocol; only the minimum number of NHP were exposed to this type of protocol because of the stress and inappetence it can cause.
- The legend for Figure 3 has been updated accordingly.

8. Figure 4 is difficult to interpret. Can the authors provide whole body images / 3D images and

objective measures from relevant regions of interest?

- Thank you for this invaluable feedback – on reflection, we can see that this is unclear. We have therefore completely redrawn Figure 4 by showing each of the four animals for which contemporaneous FDG and FDT scans were performed.
- We have also added representative SUVmax values for specific lesions to make this easier to evaluate.
- In addition, we have added the following text (starting at line 399): “**Figure 4A** shows an example where the apical and lower lesions labeled by [¹⁸F]FDG are clearly non-overlapping with those that label most strongly with [¹⁸F]FDT. In other cases (see **Figure 4B** and **C**) where there were fewer lesions, these appeared to be similarly labeled with both probes, although FDG was consistently stronger (note scale bar is an MIP of 6 for [¹⁸F]FDG and 3 for [¹⁸F]FDT). In **Figure 4D**, although there are only 5 days between the scans the disease has progressed significantly, and new lesions appear to label most intensely with [¹⁸F]FDT, suggesting that these represent areas of active bacterial replication. Further studies will be required to explore potential links between pathology and bacterial burden; these animals were instead used for exploring treatment response and meaningful necropsy samples were therefore unavailable.” and altered the legend for Figure 4, accordingly.

9. The differences in the [18F]-FDT and [18F]-FDG PET images are difficult to interpret visually in Figure 5 (panels A-B, E-F). Also, the final bacterial burden is provided for the lesions, but what is the expected bacterial burden at the start of treatments Given that individual lesions within the same animal can have highly variable PET activity as well as bacterial burden (some can be sterile), it would be important to know the number of animals imaged and a large sample size is needed to demonstrate the effect of treatment due to this variability.

- In an untreated marmoset, the bacterial burden in lesions can range from 3 to 7 log₁₀ depending on the type of lesion (cavity, necrotic, fibrotic, and consolidation as examples), the duration of the infection, and the volume of the lesion. Marmosets are put on treatment at about 7 weeks of infection.
- Sterile lesions are rarely found in *Mtb*-infected marmosets unless they have been on treatment for at least 6 weeks (see, e.g., Antimicrob Agents Chemother. 2015, 59, 4181).
- Three animals were imaged with FDT and FDG and then treated with the HRZE regimen and imaged again. Figure 5 shows one of these animals and the change in signal of the pulmonary lesions with treatment.
- Figure 5 has been updated to show 3D renderings as well as transverse and sagittal views of the lesions and the figure legend has been updated accordingly.

10. In figure 6, [18F]-FDT PET (panel A) shows high uptake in heart and the liver. Why is that? Conversely, [18F]-FDG PET shows much lower activity in the heart and liver, which is unusual as the myocardium takes up [18F]-FDG. Can the authors explain these data?

- The overall SUV range for [¹⁸F]FDT is lower than for [¹⁸F]FDG, and so reflected by different scales. This has been clarified in the legend for Figure 6.
- The signal to noise ratio for [¹⁸F]FDT is lower than it is for [¹⁸F]FDG, and so residual [¹⁸F]FDG is observed in the heart and liver of the primates.
- It should also be noted that [¹⁸F]FDG uptake in the heart is variable in species. In humans uptake can be high in one imaging session and low in the next, unrelated to serum blood glucose levels or

BMI (see, e.g., J. Clin. Imaging, 2022, 12, 37; PLoS One. 2018, 13, e0193140; J. Nucl. Cardiol, 2020, 27, 1296).

- Cynomolgus macaques (Figure 6) and the marmosets (Figures 3-5) are both fasted and sedated prior to radiotracer injection but myocardium uptake is not predictable in each scan or each animal.
- A statement and reference to this variable effect has been added to Methods (line 916).

11. Can the authors specify the number of animals used for the studies shown in Figure 7. In fact, all figure legends should clarify the number of animals used for each study, which is currently unclear. For some studies the sample size is small (e.g. n = 2) and therefore it is difficult to draw definitive conclusions.

- We thank the reviewer for pointing out our error.
- Correcting text, adding the number of animals has now been added to the legends and to the Methods (line 839): “Two uninfected and eight infected marmosets on protocol LCIM-9 were imaged with [¹⁸F]FDT one or more times as well as contributing to other experiments. The marmosets were pair-housed in an approved ABSL3 facility at NIH. Three rhesus macaques from the NIH PET center were imaged dynamically with FDT under protocol PET-14-01. FDT studies in three cynomolgus macaque at University of Pittsburgh were approved by its IACUC and Division of Radiation Safety and all MTB infected animals were pair-housed in an approved ABSL3 facility (Regional Biocontainment Facility, Pittsburgh PA).”
- As noted above, the numbers are small because the experiments were conducted in *Mtb*-infected primates with active disease that were also on other protocols with predetermined experimental endpoints.
- In addition, only 1 macaque or 2 marmosets could be imaged each day because of the size of the animals and number of PET bed positions needed to capture the lungs of the animals in a BSL-3 vivarium workday.
- In Figure 7A we show sequential images over time from 1 monkey and the data in Fig 7B come from that monkey.
- These details have been added to the legend of Figure 7.

Other comments:

12. Reference 1 can be updated to the 2022 report.

- Thank you – the most current reference (now 2023) has been inserted.

13. Page 18, second para: “‘pre-blocking’ experiments were conducted with both trehalose and ‘cold’ [¹⁹F]FDT (administered 1 h and 5 minutes)”. Based on the figure 3 legend it seems that the ‘cold’ [¹⁹F]FDT was administered at 1 h as well as 5 minutes prior to the PET studies, but this is unclear in the text. Please clarify why blocking agent was administered twice, which should also be clarified in the results section.

- Methods used to demonstrate blocking of probe binding vary greatly in the literature as to the amount of excess blocker applied and when it is administered. Blocking agent [¹⁹F]FDT was used here administered twice at about 100 x the anticipated radioactive compound dose in a manner estimated to be consistent with turnover by the enzymatic target. Recall that this is not a typical receptor block (see also discussion above).
- We have clarified this in the legend to Figure 3 and Methods (line 904).

14. *What is the expected limit of detection of this PET imaging approach?*

- Sterile lesions have the lowest total SUVbw, while lesions with higher burdens show higher FDT accumulation. For example, in figure 3i, there are two lesions at the origin that had no colony growth, and very low Total SUVbw (21 and 25). From these we can estimate > 30 SUVbw.
- Indeed, the next lowest measure in our data set is a 95 Total SUVbw with 0.9 log CFU.
- FDT uptake is directly related to bacterial number (Figures 3 and 6).
- At this stage, more data will be needed to set a more precise detection limit, in part because lesion size also varies and lesion volume is not directly related to bacterial content, especially in treated animals.

15. *Could the authors provide a summary of the primary data (clinical pathology / lab data, etc.) for the toxicology studies?*

- Thank you for this suggestion. We have revised and expanded the description in the Results section to the following (line 471): “Finally, single (acute) and multiple (chronic) intravenous (iv) dose toxicity studies of [^{19}F]FDT, enabled by ready synthesis (see above), were conducted in both rats and beagle dogs (males and females both) via a contract with SRI International (Menlo Park, CA). The animals were given either daily iv injections of [^{19}F]FDT at $100 \times$ the expected human dose for seven consecutive days or a single iv injection at either $100 \times$ or $1000 \times$ the expected human dose once (see study schema tables in the methods). Mortality and morbidity, clinical observations, body weights, food consumption, hematology, serum chemistry, and coagulation parameters (beagles only), organ weights, and gross pathology / histopathology were evaluated daily for 9 days (acute toxicity) or 21 days (recovery group). There were no adverse findings in any parameters measured in the studies that were outside the expected range of normal. The no observed adverse effect level (NOAEL) in Sprague Dawley rats was at least 13.2 mg/kg when given as a single IV injection or 1.32 mg/kg/day when given by daily IV administration for 7 consecutive days and the maximum tolerated dose (MTD) was not reached. The NOAEL in beagle dogs was at least 4 mg/kg when given as a single IV injection or 0.4mg/kg/day when given by daily IV administration for 7 consecutive days and the MTD was not determined. All animals survived until scheduled euthanasia (day 9 or day 21) except for one rat with tail lesions that were judged to not be test article related and that was euthanized earlier.”
- If desired, full study reports could be provided as supplements to the manuscript.

Reviewer #2:

*Khan et al report the synthesis and characterization of ^{18}F -labeled trehalose (^{18}F -FDT) as a candidate for PET reporting of *Mycobacterium tuberculosis* (Mtb) infection in vivo. The importance of such a technology, should it be clinically successful, is clear: tuberculosis is a disease that afflicts 10 million people and kills 1.5 million people on an annual basis, and there is not currently a way to accurately and non-invasively image active disease progression and response to therapy in infected individuals. Although the standard PET tracer ^{18}F -FDG can be used for TB imaging, it reports on host inflammation rather than metabolically active Mtb, and this approach can be confounded by non-TB lung pathologies. The idea here is to develop a TB-selective tracer based on ^{18}F -labeled trehalose. Trehalose is a disaccharide that Mtb enzymatically incorporates into surface lipids. The idea is founded on a strong body of literature*

demonstrating that various trehalose derivatives can enzymatically incorporate into Mtb.

This is a sweeping, tour de force study inclusive of synthesis, enzymology, animal models, disease imaging, and tracer production/safety. The scope of the study is remarkable and its potential impact is exceptional, as noted above. Most of the key assertions about this technology are supported by the data presented.

- Thank you.

There are some issues described below that need to be addressed. Comments are given below in categories of tracer synthesis, labeling characterization, and imaging/safety studies, and additional minor comments are given at the end.

Synthesis

Chemical/chemoenzymatic synthesis routes to four different FDT isomers with fluorine at different positions were developed to enable a comprehensive analysis and selection of the most promising PET tracer candidate. This approach proved useful, as careful analyses revealed that one of the isomers could not be efficiently accessed chemoenzymatically by the reported method (3-FDT) and two were prone to degradation in serum or in vivo (6-FDT and epi-4-FDT). 2-FDT was identified as the most promising isomer. A major strength of the work is that 2-FDT can be accessed directly from the standard PET tracer FDG. This was done using a robust one-pot enzymatic synthesis process employing hexokinase, OtsA, and OtsB enzymes that were produced in high yield without endotoxin present. It was clearly demonstrated that the carefully optimized method could generate very large amounts of the cold analog 19F-FDT and reliably produce the hot analog 18F-FDT from FDG in good radiochemical yield (~40%) and high purity. The characterization of FDT structure and purity was impeccably done.

There are several issues with the synthetic work that need to be addressed:

1. Four different enzymatic systems were tested for synthesis of FDT starting from FDG, including the three-enzyme system noted, a two-enzyme system using an OtsAB fusion protein, and two different TreT enzymes which can convert FDG to FDT in a single step. Various parameters were assessed to decide which system to proceed with. One issue was that the enzyme kinetics experiments (Table 1) measured only native glucose/G6P substrates as the acceptors, rather than FDG/FDG6P that would be used in FDT synthesis. Thus, the data do not establish which of these systems has the best activity for FDT synthesis. Kinetic analysis of FDG (most importantly 2-FDG) would need to be done in order to provide the most relevant comparison, otherwise the limitation of the experiments done should be noted.

- The reviewer is absolutely correct and, as suggested, we have altered the text (line 209 and line 211 onwards) to make it very clear that we are using these kinetic parameters only as a proxy under conditions that we identified and not as full guide to the efficiency of these systems.

*2. Related to the above, there are literature reports that used TreT enzymes from *T. tenax* and *T. uzoniensis* to synthesize cold 19F and hot 18F-FDT from FDG in one step, which should be discussed (PMIDs: 27560008, 30428395, 32953231). This includes a one-step one-enzyme radiosynthesis of FDT from FDG done by TreT from *T. tenax* in 30 min and 70% radiochemical yield (PMID 30428395). In addition, the TreT enzymes can efficiently make 3-position and 6-position FDT and the *T. uzoniensis* enzyme can make the 4-position FDT (PMID: 27560008 and*

32953231), so the isomer versatility is good. Therefore, the three-enzyme system using OtsAB benefits from excellent pyrogen-free enzyme production yields and the ability to scale up FDT synthesis. On the other hand, the one-enzyme TreT systems benefit from one-step synthesis, isomer flexibility, and efficient FDT syntheses, but the enzyme production yields are lower. A more thorough and balanced consideration of methods and their advantages and disadvantages is warranted.

- We thank the reviewer for these additional references, which are indeed excellent examples of alternative systems – we do not seek to discount any. These are elegant and complementary methods that could indeed provide future pyrogen-free methods.
- We have adjusted the text accordingly to add these citations (lines 194, 229 and 541).
- We have also made clear the potential of these other systems in same strategic vein in the Discussion (line 541).

3. Figure S9 shows that the TreT enzymes had modest conversions of FDG to FDT (10-40%). However, these results are inconsistent with the literature reports from independent groups, in which quantitative conversions were obtained in ≤ 1 h (papers noted above). Compared to the literature reports, in this study $\sim 1/10$ th the TreT enzyme was used (Table S6) and the reactions were run at sub-optimal temperatures (37 C) (Page 18 of SI). These enzymes are thermostable and reported reaction temperatures are rather run at 60-70 C, at which enzyme activities are much higher (PMID: 27560008). This is likely why the results were inconsistent.

- We thank the reviewer for this and we acknowledge the limits of our comparison. We aimed to make a comparison based on catalyst loading, with a view to scale-up (as the reviewer fairly notes), and so found the use of higher enzyme loadings (and indeed variable temperatures) problematic.
- We absolutely acknowledge that, in our hands, we were unable to further optimize, at this smaller scale of enzyme use, the utility of other systems and do not at all deny their potential (see also point added to the Discussion).
- To be clear, we do not attribute this apparent difference to non-reproducibility in any way – these are different requirements and conditions and have now added further text to aim to make this absolutely clear to the reader also (line 227).

4. Given the above points, the conclusions about the synthesis (“higher biocatalyst expression yields, flexibility of enzyme usage (differential loadings), stability and superior kinetic parameters, which, together, gave rise to more efficient synthesis in our hands”) should be revisited.

- Thank you – yes, indeed – we agree that this comparison is not necessary and lacks sufficient objectivity and so have now deleted it (line 225).
- We thank the reviewer for their diplomatic suggestion and language.

Characterization of labeling

There were experiments done to characterize metabolic stability of the various FDT isomers. It is clear that 6-FDT and (to a lesser extent) epi-4-FDT have some amount of breakdown. This in part led to judicious focusing on 2-FDT, which Figure S3G shows had higher stability. However, there were some issues with the other data presented on this topic.

5. Figure S3, note that captions for panels C and D are opposite figures, please fix

- Thank you – now corrected.

6. *Figures S35 and S36 also aim to show metabolic stability of FDT. In S35, it is not clear how long and at what temperature the samples were incubated in plasma. Also, were multiple time points assessed? (note: there are no letters on the figure corresponding to the caption).*

- We apologize for the confusion / error but it should be made clear that Figure S35 was both an illustration of a method used for detection of FDT within human plasma samples from *in vivo* experiments and an assessment of immediate stability and sequestration.
- We acknowledge that our phrasing both in the title and main text was poor and have now corrected this (Figure S35 title and line 331 onwards).
- We have also now added the caption labels for Figure S35 – thank you for spotting this error.
- Assessment of FDT's metabolic stability was / is in fact given in data found in Figure S3, S36 and now also S37 (see below) – we have also now extended our assessments to longer timepoints in NHPs and mice (see below).

7. *In S36, the sample was only taken at 5 min and the measurement appears to be a qualitative assessment of whether FDT is present. It would seem difficult to conclude that there is no degradation of the tracer from this experiment. Is there a plot for 2-FDT similar to those for the 6- and epi-4 compounds in Fig S3C and D that could be shown?*

- Data for 2-FDT is given in Figure S3 in tabular form in panel G. We have now reworded the legend to make this more clear. We have also added a callout to the legend of Figure S36.
- In additional experiments we have also now also added stability data as well as clearance data in both NHPs (marmosets) as well as mice (see Figure S37). This now extends the data in Figures S3G and S36.
- In addition, we have made an initial assessment of liver metabolism using *in vitro* human microsomes (see Figure S38).
- Corresponding text has been added to the Methods section and the text in main manuscript altered accordingly.

8. *The experiments to characterize Mtb labeling by FDT had some issues that need to be addressed. Most significantly, there were no *in vitro* uptake studies done in cultured Mtb cells. This is a relatively simple and inexpensive experiment that should be a precursor to *in vivo* imaging, and it provides an opportunity to investigate uptake mechanism in a more straightforward manner. For example, in the cited study on 18F-sorbitol, *in vitro* uptake in a panel of different bacteria and mammalian cell lines was done prior to *in vivo* imaging of infection (PMID: 25338757). In the above-noted study on 18F-FDT, *in vitro* uptake in *M. smegmatis* and mammalian cell lines was done (PMID: 30428395). However, *in vitro* uptake of FDT in Mtb has not been previously demonstrated. Ideally, this experiment would be conducted, and data shown, preferably also comparing uptake in other relevant bacterial species and potentially in mammalian cell lines. FDT is described in the paper as a TB-selective tracer, but selectivity against other types of bacteria is not demonstrated. This is of high interest in establishing the utility of FDT for TB imaging, as many types of bacteria have trehalose transporters and trehalose-metabolizing enzymes.*

- As we also note above, the reviewer is correct that other organisms are capable of utilizing trehalose, however no other organism other than non-tuberculous mycobacteria suggested, is likely to concentrate ‘intact’ trehalose.
- We have previously shown that the analogue FITC-trehalose that we cite in the text is not accumulated by *Staphylococcus aureus*, *Pseudomonas aeruginosa*, or *Haemophilus influenza* (Backus et al 2011).
- It should be noted that, here, we are not proposing that FDT would be used for the sole or even primary diagnosis of *Mtb*-mediated disease; there are indeed excellent existing ways. Instead, we are proposing that ¹⁸F-FDT would be used in patients known to be infected with TB as a means of monitoring their response to therapy and perhaps in determining when sufficient treatment has been received.
- While fungal and non-tuberculous mycobacteria occasionally occur concomitant with infection with *M. tuberculosis* this is unusual and unlikely to interfere with studies designed to compare treatment regimens or establish treatment duration with an existing regimen.
- As noted above, we have altered the abstract and text accordingly to make this point clear.

9. *The possible mechanism of FDT incorporation was investigated through TLC analysis of putatively labeled lipids from ex vivo or in vivo FDT treatment experiments. The identification of 18F-labeled trehalose mycolates would support a mechanism in which FDT is lipidated by antigen 85 as shown in Figure 1. Based on the TLC analysis shown (Figure S3E and F), it is claimed that radiolabeled TMM and TDM are observed, but the data as presented do not clearly support the conclusion: The TLC lanes are not labeled so it is unclear what standards/conditions are being presented.*

10. *The annotations on the TLC plates are confusing. In some cases, the arrows do not clearly point to a spot, and for the epi-4-FTre label, is this referring to the unmodified tracer?*

11. *Why was epi-4-FDT used for these experiments instead of the 2-FDT tracer that is the main focus of the manuscript?*

12. *The main text says authentic standards were compared to assist in identifying spots of interest, but these data/comparisons are missing. It is difficult to judge whether the spots shown are likely to represent labeled lipids without standards shown.*

13. *The extraction procedure and TLC conditions were not described in detail. In the main text methods, there is a section on “detection of labeled mono and dimycolates” but it only describes analysis of blood and urine samples and does not seem relevant, whereas the Figure S3E/F caption refers to lung samples and gives very little detail. The SI methods did not appear to cover these experiments. It is very challenging to interpret the data shown without more experimental detail.*

14. *In addition, in vitro experiments to investigate mechanism of uptake would be valuable. Presumably, extraction and analysis of labeled lipids from *Mtb* grown in vitro would be a good first step, prior to going into infected animals which significantly complicates the analysis. Given that FDT incorporation into mycolates has never previously been demonstrated, this would be a good experiment, particularly in light of the at-present questionable ex vivo/in vivo lipid labeling experiments.*

- We have previously demonstrated lipidation of trehalose analogues to their corresponding mycolates (see Backus *et al.* 2011) but we note and agree with the reviewer’s reservations about the use of (and indeed the relevance) of 4-epi-FDT in the experiments discussed above.
- 4-epi-FDT behaves differently to 2-FDT and we would not wish these experiments to be overinterpreted. We have now removed these from the manuscript, given their tangential relevance. We thank the reviewer for highlighting this.

- Therefore, as suggested, we have now conducted *in vitro* experiments to address this aspect directly for FDT through the generation of ¹⁴C-labelled-[¹⁹F]FDT ([¹⁴C,¹⁹F]FDT). This revealed incorporation into less polar, lipidic species with TLC silica gel R_f values that match those of prior analyses, consistent with incorporation of FDT into FDT-MM and FDT-DM mycolates.
- These data have been incorporated into Figure S3, and the text in the manuscript altered accordingly (line 347 onwards and Methods, lines 966 onwards).

15. *Another question related to characterization of mechanism is if the authors considered whether the Mtb trehalose transporter could be involved in uptake of the tracer, as previously demonstrated in M. smegmatis (PMID: 27560008). This is of interest because if the tracer is solely processed through antigen 85 that would lend toward mycobacteria-selective uptake and a TB-selective tracer. On the other hand, if uptake is driven by trehalose transport, there could be selectivity issues since as noted above this is a common feature of many bacteria (albeit there are different transporter types with likely different specificities for trehalose derivatives). Discussion of the potential involvement of trehalose transport systems in FDT uptake would be valuable.*

- This is indeed an interesting question. As the reviewer highlights, there are additional mycobacterial pathways that internalize and/or metabolize trehalose including those that utilize the *lpqY-sugA-sugB-sugC*-derived apparatus. As yet, corresponding knockouts, and particularly complemented strains, that are found in *M. smegmatis* whilst known are less tractable in *Mtb*. Their evaluation as part of future studies would be of interest in establishing further aspects of selectivity.
- As the reviewer notes, the trans-mycolylation activity of Ag85 isoforms have been established as competent *in vitro*, indeed also with [¹⁹F]FDT (Backus et al 2011) and the kinetic model for Ag85 (Barry et al 2011, *J. Am. Chem. Soc.* **2011**, *133*, 13232) is consistent with observed plasticity.
- As suggested, the citation to the excellent work on *lpqY-sugA-sugB-sugC* ABC transporter systems, including that in *M. smegmatis*, is useful and a pointer to this and future experiments have now been added to the manuscript in the Discussion (line 537).

Imaging and safety

The PET imaging experiments, conducted in multiple animal infection models, strongly support FDT as a tracer for monitoring Mtb burden and response to antibiotic treatment in vivo, which as noted above could be a powerful preclinical and potentially clinical capability. Some questions on the imaging experiments are given below:

16. *The comparison to TB imaging using FDG demonstrated differential labeling patterns in the replicate shown. The n=3 for this experiment, can images from additional replicates be added to the SI? Also, if there were uninfected control animals for this experiment, those would be useful to show in Figure 4 and/or the SI.*

- Figure 4 has been completely redrawn to show all four animals that were imaged with both probes. See also response to Reviewer 1, Question 8.

17. *It was observed that uptake of FDT reached maximum signal to noise at 90 min, how does this compare to FDG and, if different, please comment? Was FDT/FDG uptake monitored at earlier time points than 60 min, and if so, how did overall signal and distribution compare?*

- With FDT, the minimum dwell period was 60 minutes, with 90 and 120 also collected in early experiments.
- In marmosets, the earliest FDT and FDG PET images were collected at 60 minutes. Prior unpublished experiments examined FDG uptake at several timepoints (45 min, 60 min, 90 min) and found background in the lung decreased over time. Signal-to-noise ratios improve slightly from 60 to 90 min. As the animals are generally sick, longer sedation time was therefore not considered.

18. *Figure 7B/C shows relatively little FDT in lungs, can the authors comment on the consequences of this for imaging pulmonary TB? It would also be interesting for the authors to comment on how the other FDT biodistribution data may impact potential imaging of extrapulmonary TB.*

- We thank the reviewer for the comment. In these distribution studies, the vast majority of the PET probe is expected to pass from the blood pool into the urinary tract to the urine or sometimes the hepatobiliary tract. Figure 7B summarizes the radioactivity exposure of the entire experiment shown in 7A. In the early panels of 7A the tracer passes through the lung leaving little signal in the organ thus increasing the opportunity to detect accumulation of the probe in a lesion in the lung. In figure 7C, where normal tissue pieces were harvested and the proportion of the injected dose captured was calculated, the distribution into organs was similar.
- As Referee 3 also notes, little FDT in lungs is therefore, in fact, advantageous, as there is little non-specific radiotracer background, enabling visualization of any specific uptake in TB lesions in lung: “In vivo biodistribution in marmoset shows no significant [¹⁸F]FDT uptake in normal lung tissues, radiotracer uptake in tuberculous lung lesions ...”.
- As the organ retentions of the FDT are low, extrapulmonary lesions with numerous *Mtb* bacilli might retain enough of the FDT tracer to be detectable in the PET images, but we do not have data to address this in the biodistribution experiment.

19. *The safety studies, conducted in multiple animal types and enabled by the excellent scaled up cold FDT synthesis, demonstrated no adverse effects, setting the stage for subsequent clinical testing in humans.*

- Thank you.

Additional minor comments and edits

Title: The meaning of “distributable” may be somewhat ambiguous to some readers as to whether this means the technology is distributable to (pre)clinical facilities or the tracer is distributable to different tissues in vivo

- We note the possible ambiguity but feel that in the context of the abstract (e.g. line 20) that this will become clear upon further reading.

Abstract: “can act as a mechanism-based enzyme reporter in vivo” is a bit confusing, perhaps “mechanism-based reporter of Mtb-specific enzyme activity in vivo” or something to this effect?

- Thank you. We have now altered this (line 11).

Abstract: suggest “custom-made” instead of “bespoke”

- Thanks for the suggestion, we have now altered as suggested (line 21).

Page 3: Cite WHO 2022 report/data.

- We have now updated this citation (line 27)

Page 3: The sentence beginning “The analysis⁴ and internationally-agreed, comprehensive monitoring of access to PET-CT” is difficult to follow, please rephrase.

- Thank you – yes – we have now edited this to improve the flow (line 41).

Page 4 typo: “...and so only can only help...”

- Corrected (line 72) – thank you.

Page 4: Cord factor refers to TDM, not TMM (sentence is ambiguous).

- Indeed – thank you – now corrected (line 83)

Page 4: Ag85 is generally considered to be cell envelope-associated or secreted, but to this reviewer’s knowledge is not commonly thought to, or depicted to, significantly associate with the plasma membrane as shown in Figure 1a.

- As suggested, we have adjusted Figure 1a

Page 5: “could function as both highly specific and sensitive reporters” (remove “a”)

- As suggested, now corrected (line 89).

The authors should comment on why epi-4-FDT rather than 4-FDT was targeted, e.g. synthetic accessibility or other reason.

- Yes, this was synthetic expediency and a note has now been added to the text (line 124).

Can authors comment on the low yield (15%) of fluoride substitution en route to 4-epi-FDT?

- This was essentially unoptimized as our initial scoping set out generate access to the radiotracer for functional evaluation and we did not extensively revisit the radiosynthesis.

- As the reviewer will appreciate, the synthetic expediency that is generated by ready access to trehalose as starting material is offset, in part, by the poor substitution chemistry at C-4 in this configuration (beta-oxygen effect coupled with ‘axial’ attack of nucleophile).

Page 9 define RCY (this is given later); no need to capitalize “one pot”

- Both now corrected (lines 170 and 174, respectively).

SI P56, Figure S9 caption, give FDG substrate amounts as concentrations rather than ug.

- Thank you – now corrected.

Page 16 Figure 2B, give FDG substrate in concentration

- Thank you – now corrected.

Comment on why such a large excess of UDP-Glc was required for optimal conversion (130-260 mM), and whether that led to any purification issues?

- This was more a question of pragmatically driving the kinetics in a predictable manner in a system that is essentially saturated for one of two substrates (based on estimated K_M). In our hands, this allows more ready scale-up in biotransformations by creating a pseudo-single substrate kinetic regime.
- This did not create for us any associated problems of purification.

Figure 2C, is this monitoring over time? If so, show time durations on figure.

- No – this is better considered as monitoring of each step of the one-pot method. The legend has been altered accordingly.

Page 14, “consistently and repeatedly.”

- Corrected (line 289)

- *Page 15, define GMP on first use.*

- Now added (line 299)

Page 16 Figure 2D and associated text, can the authors clarify exactly what is meant by batch? Are these just repeats of the reaction with fresh enzyme/reactants, are enzymes being re-used, are they from different protein prep batches?

- These are using fresh enzyme and reactants (sometimes from newly expressed batches). There is no re-use or recycling of enzyme in the current work.
- We have added to the text accordingly (legend of Figure 2, line 281)

Figure S33, caption says (for a-c) and for d, but it is not clear what these letters refer to.

- Thank you for spotting this – we have now corrected this legend.

Page 30 “scales of up to grams” statement clearly refers to 19F-FDT but occurs immediately after referring to 18F-FDT.

- Thank you, this was clumsy – we have now corrected the text (line 550).

Page 33, should be Michaelis-Menten

- Thank you ! – now corrected (line 598).

Since a significant number of small errors and typos were noted, the authors are strongly encouraged to carefully check and edit the main text and SI thoroughly.

- Indeed – thank you for your patience and careful noting – we believe we have now corrected all of these (highlighted).

Reviewer #3:

The authors of this interesting work investigated pathogen-specific imaging of tuberculosis with radio-fluorinated trehalose. The authors leveraged on the specificity of trehalose utilization by Mtb for specific pathogen targeting. Different fluorinated trehalose molecules were tested for their efficiency of radiosynthesis, length of reaction time, and in vivo stability. [18F]FDT demonstrated superior performance and was investigated further. A scalable technique for synthesizing [18F]FDT was subsequently described. [18F]FDT showed rapid and specific incorporation into mycobacterial mycolic acid. In vivo biodistribution in marmoset shows no significant [18F]FDT uptake in normal lung tissues, radiotracer uptake in tuberculous lung lesions with signal reduction in blocking experiment confirming the specificity of binding, minimal increase in the intensity of radiotracer uptake in lesions at 90 minutes compared to 60 minutes post tracer injection with no further improvement in avidity beyond 90 minutes, and a direct relationship between the radiotracer avidity of the lesions and the number of culturable Mtb within the lesions. The spatial distribution of FDG uptake differs from that of FDT, reflecting the differences in the target engagement between the two tracers. Interestingly, there was a significant reduction in FDT but not FDG signal corresponding to a reduction in Mtb burden in treated marmosets, suggesting that FDT may be a useful biomarker of Tb response assessment. The safety of [18F]FDT in the experimental animals was demonstrated. In dosimetry study, low FDT uptake in normal organs was reported with the target organ being the urinary bladder wall. FDG is the most commonly used radiotracer for infection imaging, including Tb imaging. Its lack of specificity for infection limits its clinical application. A need, therefore, exists for novel

radiotracers that are specific for pathogens. The report in this manuscript of 18F-FDT is a welcome development towards achieving this goal.

- Thank you.

Specific comments:

1. Results - [18F]FDT is also an Effective Mtb-Radiotracer in an 'Old World' Non-Human Primate: As shown in figure 6, [18F]FDT PET signal at 60 and 120 minutes were compared between lesions with culturable Mtb bacilli and those without. Why was 90 minutes, the optimum imaging time point left out here? Is there same data for FDG as well? It will be great to see how FDT compares with FDG in this regard.

- We agree it would have been ideal to have aligned data in both species and for both tracers, but also, we try to minimize the time that infected animals with compromised lungs are sedated. In the marmoset, 2 PET bed positions of 10 minutes each were needed to capture the distance from the neck to the tip of lungs with a several centimetres manually-programmed overlap in the collection window on the PET scanner used (Focus 220). Therefore, it was possible to collect a complete series of PET scans every 30 minutes for the marmosets.
- By contrast, the 90-minute time point was technically difficult for the cynomolgus macaques since 3 or 4 bed positions had to be collected to capture from the neck to the tips of the lung on these larger animals with the same type of scanner.
- Data collection in the two species was concurrent so we did not know at the time the optimal time point for the marmosets at the time the cynomolgus macaque scans were collected. We therefore did not collect 90-minute scans for cynomolgus macaques for comparison (and therefore whether optimal or not in this model).
- It should be noted that cynomolgus macaques show slightly slower metabolism; 60 minutes was previously established as giving a good signal-to-noise ratio for FDG in the two NHP species.
- In general, limited longer FDG scans have been collected by our groups to protect the welfare of NHPs.

2. An important finding from this study is the spatial incongruence in the uptake of FDT versus FDG in the same lesion. This may support differences in target engagement between the two tracers, however, a proof of this is necessary. This proof needs to show the concentration of Mtb bacilli in the region of FDT uptake and the concentration of inflammatory cells in the region of FDG uptake.

- This is a very interesting idea, and one that would be worth pursuing in future experiments, where perhaps use of immunofluorescent staining of immune cell types with 'RNA-scope' probes of *Mtb* in specifically cut sections might prove effective.
- This would require additional NHPs to be dedicated to this experiment and an associated future protocol.
- As shown in Figure 4, the regions with more intense FDT signal are closely positioned or intertwined with the regions of higher FDG uptake in some lesions, so our typical lesion by lesion necropsy would be unlikely to provide this information.

3. An important drawback of FDG for Tb response assessment is the ability of dead bacilli in treated Tb lesion to induce inflammatory changes causing FDG uptake in sterile lesions. FDT, as

described here, has the potential to address this limitation. It will, therefore, be interesting if the authors could perform additional experiments to validate this.

- We wholeheartedly agree. Our first attempts at this are the chemotherapy studies that we present here. Potent treatment showed loss of the FDT signal but not FDG.
- We plan to conduct future additional studies to verify the lack of labelling of dead bacilli using other candidates.

4. Tb lesions in humans are highly diverse. The animal works presented here investigated the performance of FDT PET imaging in a limited spectrum of Tb lesions. In the comparative images of FDT and FDG PET as shown in figures 4 and 6, the signal from FDT imaging appears much less compared with FDG. This is, therefore, a concern regarding the performance of this novel tracer in detecting small lesions, especially in the context of the significant partial volume averaging that occurs with respiratory motion. Cavitory lesions are of particularly clinical interest, and the performance of FDT in this disease phenotype is unknown.

- In detecting signals in small lesions, many of the marmoset and cynomolgus lesions measured in this work were, in fact, small (only 3 to 5 mm in size), while most human lesions are, of course, much larger.
- Partial volume averaging will therefore be better in human subjects where this probe will ultimately be tested.
- We agree with the reviewer that cavities are of particular interest and in our limited pilot experiments we have imaged only a few animals.

5. Discussion: Please make a brief mention of the comparison in organ dose from FDT as reported here versus FDG as reported in the literature.

- Using the corresponding FDG doses (5 mCi), the exposure for the top 3 FDT organs would be:

Bladder	FDT 0.143 mSv/MBq	vs.	FDG 0.086 mSv/MBq
Kidney	FDT 0.119 mSv/MBq	vs.	FDG 0.020 mSv/MBq
Adrenals	FDT 0.022 mSv/MBq	vs.	FDG 0.013 mSv/MBq

Note that we used a 2 hr bladder void interval, whereas the FDG table uses a 1.5 hr void. Our doses would be somewhat less with a 1.5 hr void interval. Thus, there is a 2-to-5-fold difference in exposure.

- This has been added to the manuscript (line 463).

Reviewer #1 (Remarks to the Author):

This revised manuscript utilises chemoenzymatic methods to synthesise 2-[18F]fluoro-2-deoxytrehalose ([18F]-FDT) using 2-[18F]fluoro-2-deoxy-D-glucose ([18F]FDG) as the substrate. Changes to the prior manuscript are appreciated and additional comments are below:

1. Thank you for providing the specific activity for FDT. While in-house FDG production with higher specific activities may solve some of these problems, it may somewhat limit the wide use of FDT, as many commercial vendors (globally) provide low specific activity FDG, which will be the precursor for FDT. It is therefore suggested that the abstract "could now usher in global, democratized access to a TB-specific PET tracer" and the manuscript be toned down to reflect this limitation in the methodology to produce high-specific activity FDT or the amounts needed (10-20 mCi) for human use.

2. In the abstract, please change, "Mtb-specific" and "TB-specific" to "bacteria-specific". This is also needed throughout the manuscript.

3. Human liver microsomal assays and whole-body images are appreciated.

4. The Weinstein et al. *Sci Transl Med.* 2014, describing FDS did demonstrate specificity using several controls, sterile inflammation (Fig. 2), brain tumour (Fig. 3) and other bacteria (Fig. 4). Neither Ordonez et al. *Antimicrob Agents Chemother.* 2012 nor Ordonez et al. *Antimicrob Agents Chemother.* 2015 are describing a bacteria-specific tracer and rather assessing antimicrobial PK. Therefore, bacterial specificity continues to remain an issue in the current manuscript and I continue to believe that sterile inflammatory controls are critical to demonstrate that FDT is bacteria-specific (see also point 3 above). For context, FDG (which is not bacteria-specific) is already an excellent PET tracer to monitor disease TB in animals and patient (Chen et al. *Sci Transl Med.* 2014; Coleman et al. *Sci Transl Med.* 2014; Xie et al. *Sci Transl Med.* 2014). While in vitro studies are very valuable to determine specificity, for in vivo studies there is an additional concern related to non-specific tracer accumulation at infection sites (which are also inflamed and thus blood vessels are leaky). Please also see point 4 above that suggests that there may be free FDG (during FDT imaging) or that FDT is being taken up by the myocardium, which makes it even more important to test for specificity. Therefore, studies are needed to ensure that the tracer accumulation in the animal studies is specific. The reviewer would also suggest dynamic PET that can help assess whether the uptake is specific or not.

5. Blocking studies: The reviewer appreciates the challenges of the work with non-human primates, and the explaining for only moderate level of blocking. However, a sample size of two remains limited, given the lack of other data to demonstrate specificity. Additionally, the reviewer analysed the data from the studies quoted by the authors in support of low numbers for the NHP studies but the number of animals used for the blocking studies is either not clear (*J Nucl Med*, 2022, 63, 1912, methods state that 3 NHPs were used) or the studies provided data from other species (rodents and / or humans with autoradiography) and / or dynamic PET (*Acta Pharm Sin B*, 2023, 13, 213, *Pharmacol. Res.*, 2023, 189, 106681, *Nucl Med Biol*, 2007, 34, 153). Dynamic PET helps with establishing specific uptake. Given the difficulties of imaging in NHPs, can the authors provide additional data in rodent models (mouse, etc.) with a sufficient sample size to support these data?

6. The author appreciates the updated figures but it is still difficult to ascertain the differences between FDG and FDT PET. Both tracers seem to light up essentially the same areas. The reviewer respectively disagrees that sterile lesions are rarely found in Mtb-infected NHPs without treatment. Please see *Antimicrob Agents Chemother.* 2013 Sep; 57(9): 4237-4244 where sterile lesions were reported in untreated NHPs.

7. As noted earlier, the FDT PET images show substantial uptake by the myocardium, which is similar or higher than in the TB-lesions. Given the low overall SUV activity with FDT, a concern would be that even a small amount of residual FDG (either not fully converted to FDT or as a byproduct of FDT metabolism) would lead to this uptake pattern (see also points 5-7 above, about the need for establishing specificity).

Reviewer #2 (Remarks to the Author):

The authors have addressed the comments satisfactorily. I have a few remaining comments that are related to the issue of FDT specificity, some discussion of which should be added.

- In response to similar comments by reviewers 1 and 2, it is stated that “no other organism other than non-tuberculous mycobacteria, is likely to concentrate ‘intact’ trehalose.” However, this is not strongly supported up by available data. The authors have not conducted in vitro uptake studies as suggested by reviewers 1 and 2, and a recent paper that did do these studies with 18F-FDT showed uptake by various types of bacteria (PMID: 37535945, Figure S3). Many bacteria have trehalose transport machinery for intact trehalose or the products of cell membrane-associated trehalose hydrolysis (e.g., PMID: 32862781), both of which could concentrate signal in non-mycobacterial organisms. It is also not clear what the “intact” trehalose distinction would mean, practically speaking.

- As noted above, during the revision period, a new paper on 18F-disaccharide synthesis and bacterial infection imaging was reported (PMID: 37535945). This includes 18F-FDT synthesis using trehalose phosphorylase and in vitro uptake studies that demonstrate various bacteria accumulate FDT. This new work does not diminish the importance of the manuscript under consideration. However, it should be cited and mentioned.

- In multiple instances, comments on 18F-FDT specificity were made on the basis of prior experiments using a trehalose-fluorophore conjugate FITC-trehalose. Comparing FITC-Tre to 18F-FDT is not necessarily appropriate since these molecules may have different uptake pathways/efficiencies due to the significant structural differences (e.g., prior work shows small modifications like F or N3 are tolerated by transporter, but larger modifications like fluorophores are not and proceed via Ag85 incorporation). This caveat should be included in any comparison of the two molecules.

Finally, I noticed that in response to reviewer 1, the authors reference new Supplementary Figures S37G,H but I did not see these panels in the revised SI file.

Reviewer #3 (Remarks to the Author):

Thank you for your response to my comments. I am satisfied with the explanation provided and I have no further comments.

Reviewer #1:

This revised manuscript utilises chemoenzymatic methods to synthesise 2-[18F]fluoro-2-deoxytrehalose ([18F]-FDT) using 2-[18F]fluoro-2-deoxy-D-glucose ([18F]FDG) as the substrate. Changes to the prior manuscript are appreciated and additional comments are below:

1. Thank you for providing the specific activity for FDT. While in-house FDG production with higher specific activities may solve some of these problems, it may somewhat limit the wide use of FDT, as many commercial vendors (globally) provide low specific activity FDG, which will be the precursor for FDT. It is therefore suggested that the abstract “could now usher in global, democratized access to a TB-specific PET tracer” and the manuscript be toned down to reflect this limitation in the methodology to produce high-specific activity FDT or the amounts needed (10-20 mCi) for human use.

- We think it would be unwise to generalise on what molar activities may be available. As the reviewer will know, this can be not only site but even run variable from a given facility.
- Nonetheless, as we noted in our prior replies, suppliers can provide activities within the ranges needed.
- We therefore thank the reviewer for this suggestion but we would argue that the principle summarised in this phrase remains intact.

2. In the abstract, please change, “Mtb-specific” and “TB-specific” to “bacteria-specific”. This is also needed throughout the manuscript.

- We refer the referee to our prior replies. In our previous corrections, we altered, where appropriate, our phrasing to highlight the mycobacterial-selectivity of FDT.
- As we also noted in our prior replies, we have co-opted a pathway that is specific to Mtb [as stated in the abstract and elsewhere, p4] – these phrases therefore remain correct.
- Nonetheless to avoid ambiguity in interpretation we have, as suggested, further altered phrases describing the pathway to make this even more clear.
- As we also noted in our prior replies, whilst we see no indication of non-specific uptake, we cannot exhaustively discount this possibility – the essential point raised by the referee. We argue that this is unlikely, yet we have tempered our observations accordingly in response to the referee’s point. Nonetheless, the potential for TB-specific tracer usage remains.

3. Human liver microsomal assays and whole-body images are appreciated.

- Thank you.

4. The Weinstein et al. Sci Transl Med. 2014, describing FDS did demonstrate specificity using several controls, sterile inflammation (Fig. 2), brain tumour (Fig. 3) and other bacteria (Fig. 4). Neither Ordonez et al. Antimicrob Agents Chemother. 2012 nor Ordonez et al. Antimicrob Agents Chemother. 2015 are describing a bacteria-specific tracer and rather assessing antimicrobial PK. Therefore, bacterial specificity continues to remain an issue in the current manuscript and I continue to believe that sterile inflammatory controls are critical to demonstrate that FDT is bacteria-specific (see also point 3 above). For context, FDG (which is not bacteria-specific) is already an excellent PET tracer to monitor disease TB in animals and patient (Chen et al. Sci Transl Med. 2014; Coleman et al. Sci Transl Med. 2014; Xie et al. Sci Transl Med. 2014). While in vitro studies are very valuable to determine specificity, for in vivo studies there is an additional concern related to non-specific tracer accumulation at infection sites (which are also inflamed and thus blood vessels are leaky). Please also see point 4 above that suggests that there may be free FDG (during FDT imaging) or that FDT is being taken up by the myocardium, which makes it even more important to test for specificity. Therefore, studies are needed to ensure that the tracer accumulation in the animal studies is specific. The reviewer would also suggest dynamic PET that can help assess whether the uptake is specific or not.

- The reviewer has re-posed their prior question and so we refer them to our prior answer and the revisions and additions made to the manuscript in response.
- As we noted before, there is, of course, a delicate balance in committing to the terminal use of NHPs *versus* what will be learned.
- The added suggestion now of dynamic PET experiments is an interesting one that we agree could valuable form part of future studies.
- We note also the requirements that will be addressed by assessment for entry into Phase 1.
- Please note that we could place into context the phrase “*Please also see point 4.*”, as this is Point 4. We hope we have addressed the concerns raised.

5. Blocking studies: The reviewer appreciates the challenges of the work with non-human primates, and the explaining for only moderate level of blocking. However, a sample size of two remains limited, given the lack of other data to demonstrate specificity. Additionally, the reviewer analysed the data from the studies quoted by the authors in support of low numbers for the NHP studies but

the number of animals used for the blocking studies is either not clear (J Nucl Med, 2022, 63, 1912, methods state that 3 NHPs were used) or the studies provided data from other species (rodents and / or humans with autoradiography) and / or dynamic PET (Acta Pharm Sin B, 2023, 13, 213, Pharmacol. Res., 2023, 189, 106681, Nucl Med Biol, 2007, 34, 153). Dynamic PET helps with establishing specific uptake. Given the difficulties of imaging in NHPs, can the authors provide additional data in rodent models (mouse, etc.) with a sufficient sample size to support these data?

- We thank the reviewer for acknowledging the reasons put forward for the results obtained in blocking experiments and also for acknowledging, again, the delicate balance in committing to the terminal use of NHPs *versus* what will be learned.
- The papers we cited were intended to be illustrative of diverse blocking studies where significant insight was gained from similarly limited sample sizes, and, indeed, where such a balance was also being struck.
- We are therefore somewhat unclear what the basis is for requesting additional animal experiments, beyond the desire for greater sample size.
- We hope that our introduction and the cited examples therein were clear in highlighting the limited relevance of rodent *Mtb* infection models for preclinical imaging.

6. The author appreciates the updated figures but it is still difficult to ascertain the differences between FDG and FDT PET. Both tracers seem to light up essentially the same areas. The reviewer respectively disagrees that sterile lesions are rarely found in Mtb-infected NHPs without treatment. Please see Antimicrob Agents Chemother. 2013 Sep; 57(9): 4237–4244 where sterile lesions were reported in untreated NHPs.

- Whilst the differences are clear not only upon quantitation but also by eye, we do understand that visualization in this format is associated with the need for visual interpretation of this subtlety. We of course would not wish to overclaim on this point and we have adjusted the text accordingly to further reflect the referee's input. Thank you.

7. As noted earlier, the FDT PET images show substantial uptake by the myocardium, which is similar or higher than in the TB-lesions. Given the low overall SUV activity with FDT, a concern would be that even a small amount of residual FDG (either not fully converted to FDT or as a byproduct of FDT metabolism) would lead to this uptake pattern (see also points 5-7 above, about the need for establishing specificity).

- The reviewer has again re-posed a prior question and so we refer them to our prior answer and the revisions and additions made to the manuscript in response.
- As we stated previously, in the Figure 6 imaging that the reviewer is referring to, the overall SUV range for [¹⁸F]FDT was lower than for [¹⁸F]FDG, and so reflected by different scales.
- The critical point is that even for [¹⁸F]FDG uptake in the heart is variable from species-to-species and subject-to-subject. In humans uptake can be high in one imaging session and low in the next, unrelated to serum blood glucose levels or BMI (*PLoS One* 2018 13, e0193140). Similarity Cynomolgus macaques (Figure 6) and the marmosets (Figures 3-5) are both fasted and sedated prior to radiotracer injection but myocardium uptake is not predictable in each scan or each animal.
- We therefore suggest that it would be unfair and incorrect to speculate on origins of this signal given this inherent variability.

- The reviewer, for instance, speculates that there is residual impurity from the synthetic process used and/or from degradation. We refer them again to the sections in which we detailed the high level of FDT purity that arises in part from a 3-step (and so more ‘synthetically distant’ in conversion from FDG) process as well as the resistance to degradation that is engendered by the 2-fluoro-2-deoxy modification (as well as our experimental assessment of the resulting stability).

Reviewer #2:

The authors have addressed the comments satisfactorily. I have a few remaining comments that are related to the issue of FDT specificity, some discussion of which should be added.

In response to similar comments by reviewers 1 and 2, it is stated that “no other organism other than non-tuberculous mycobacteria, is likely to concentrate ‘intact’ trehalose.” However, this is not strongly supported up by available data. The authors have not conducted in vitro uptake studies as suggested by reviewers 1 and 2, and a recent paper that did do these studies with 18F-FDT showed uptake by various types of bacteria (PMID: 37535945, Figure S3).

Many bacteria have trehalose transport machinery for intact trehalose or the products of cell membrane-associated trehalose hydrolysis (e.g., PMID: 32862781), both of which could concentrate signal in non-mycobacterial organisms. It is also not clear what the “intact” trehalose distinction would mean, practically speaking.

As noted above, during the revision period, a new paper on 18F-disaccharide synthesis and bacterial infection imaging was reported (PMID: 37535945). This includes 18F-FDT synthesis using trehalose phosphorylase and in vitro uptake studies that demonstrate various bacteria accumulate FDT. This new work does not diminish the importance of the manuscript under consideration. However, it should be cited and mentioned.

- The referee raises the excellent work of Wilson *et al*, which we have cited. Indeed, reciprocally, Professor Wilson contacted us at the time (April 2023) that we pre-printed our work [bioRxiv 2023.04.03.535218; doi: <https://doi.org/10.1101/2023.04.03.535218>] indicating that he wished to cite it and we had a very productive and cooperative discussion. We thank him for his supportive interest.
- The radiotracer association data measured in Figure S3 of Wilson *et al* is important to be clear about. It describes the *in vitro* association [‘incorporation’] of tracer (synthesized by a one-step method) with cultures of *S. aureus*, *L. monocytogenes*, *E. faecalis*, *K. pneumoniae*, *E. coli*, *P. aeruginosa*, *A. baumannii*, *S. typhimurium*, *P. mirabilis*, *E. cloacae*. Critically, in this protocol no heat treatment nor blocking comparison is made making it not possible to conclude the significance of biological uptake (if any) as might be indicated by increase beyond background surface adsorption following attempted washing of residual tracer from culture. Wilson *et al*, correctly and cautiously describe this only as ‘incorporation’ in their text without suggesting that this is uptake derived.
- Additionally, although not specifically stated for this data, we presume that the culture incubation period for these experiments was 90 min at 37°C by extrapolation from the method given for other tracers in that work. This significant incubation period creates the strong potential for processing by bacterial species into, for example, trehalose-6-phosphate and then to glucose. This is a known inducible process in bacterial culture (e.g. *E. coli* *Arch. Microbiol.*, **1984**, 137, 70-73). Furthermore, as Wilson *et al* show, any FDG produced by such inducible in culture degradation would then be taken up by bacterial species. No timecourse is measured nor comparison made with under the same conditions with *Mtb* so an assessment of relative selectivity is not possible.

- We note also that *S. aureus*, an organism that we previously found to be incapable of trehalose analogue uptake by the pathway we exploit under conditions where *Mtb* was functional (Backus et al, *Nat Chem Biol* **2011**), is suggested by the data given by Wilson *et al.* to display essentially similar ‘incorporation’ levels to *E. coli*. This seems to further suggest that other mechanisms may be in play and mechanistic extrapolation unwarranted.
- Thus, taken together, we consider that it would be unfair to overinterpret the data of others. We note too that Wilson *et al* do not draw wide-ranging conclusions either, restricting their comments simply to the observation of ‘significant incorporation by *E. coli*’ without speculating on uptake or its origins.
- Most pertinently, for these reasons and as we have noted previously, the conditions found in such culture incubations can be more limited in their value and are very far from the *in vivo* evaluations that we explore here.

• In multiple instances, comments on 18F-FDT specificity were made on the basis of prior experiments using a trehalose-fluorophore conjugate FITC-trehalose. Comparing FITC-Tre to 18F-FDT is not necessarily appropriate since these molecules may have different uptake pathways/efficiencies due to the significant structural differences (e.g., prior work shows small modifications like F or N3 are tolerated by transporter, but larger modifications like fluorophores are not and proceed via Ag85 incorporation). This caveat should be included in any comparison of the two molecules.

- Thank you for this important note. Yes indeed – we have added an additional caveat to the text.

Finally, I noticed that in response to reviewer 1, the authors reference new Supplementary Figures S37G,H but I did not see these panels in the revised SI file.

- Apologies; these were shown as Figures S38A,B.

Reviewer #3:

Thank you for your response to my comments. I am satisfied with the explanation provided and I have no further comments.

- Thank you.
-